# Single-neuron representation of learned complex sounds in the auditory cortex

Meng Wang[1,2,10], Xiang Liao [3,10 ✉], Ruijie Li[1,10], Shanshan Liang[1,10], Ran Ding[1], Jingcheng Li[1], Jianxiong Zhang[1], Wenjing He[1], Ke Liu[1], Junxia Pan[1], Zhikai Zhao[1], Tong Li[1], Kuan Zhang[1], Xingyi Li[1,3], Jing Lyu[4], Zhenqiao Zhou[4], Zsuzsanna Varga[5], Yuanyuan Mi[3], Yi Zhou [6], Junan Yan [6], Shaoqun Zeng[2], Jian K. Liu[7], Arthur Konnerth[5], Israel Nelken [8], Hongbo Jia [4,5,6 ✉] & Xiaowei Chen [1,9 ✉]

The sensory responses of cortical neuronal populations following training have been extensively studied. However, the spike firing properties of individual cortical neurons following training remain unknown. Here, we have combined two-photon $Ca^{2+}$ imaging and single-cell electrophysiology in awake behaving mice following auditory associative training. We find a sparse set (~5%) of layer 2/3 neurons in the primary auditory cortex, each of which reliably exhibits high-rate prolonged burst firing responses to the trained sound. Such bursts are largely absent in the auditory cortex of untrained mice. Strikingly, in mice trained with different multitone chords, we discover distinct subsets of neurons that exhibit bursting responses specifically to a chord but neither to any constituent tone nor to the other chord. Thus, our results demonstrate an integrated representation of learned complex sounds in a small subset of cortical neurons.

[1] Brain Research Center and State Key Laboratory of Trauma, Burns, and Combined Injury, Third Military Medical University, Chongqing 400038, China. [2] Britton Chance Center for Biomedical Photonics, Wuhan National Laboratory for Optoelectronics-Huazhong University of Science and Technology, Department of Biomedical Engineering, Key Laboratory for Biomedical Photonics of Ministry of Education, Huazhong University of Science and Technology, Wuhan 430074, China. [3] Center for Neurointelligence, School of Medicine, Chongqing University, Chongqing 400030, China. [4] Brain Research Instrument Innovation Center, Suzhou Institute of Biomedical Engineering and Technology, Chinese Academy of Sciences, Suzhou 215163, China. [5] Institute of Neuroscience and the SyNergy Cluster, Technical University Munich, 80802 Munich, Germany. [6] Advanced Institute for Brain and Intelligence, Guangxi University, Nanning 530004, China. [7] Centre for Systems Neuroscience, Department of Neuroscience, Psychology and Behaviour, University of Leicester, Leicester LE1 7RH, UK. [8] The Edmond and Lily Safra Center for Brain Sciences, and the Department of Neurobiology, Silberman Institute of Life Sciences, Hebrew University, Jerusalem 91904, Israel. [9] CAS Center for Excellence in Brain Science and Intelligence Technology, Shanghai Institutes for Biological Sciences, Chinese Academy of Sciences, Shanghai 200031, China. [10] These authors contributed equally: Meng Wang, Xiang Liao, Ruijie Li, Shanshan Liang. ✉email: xiang.liao@cqu.edu.cn; jiahb@sibet.ac.cn; xiaowei_chen@tmmu.edu.cn

For each sensory modality, e.g., vision, audition, and somato-sensation, the corresponding primary sensory cortical region of the mammalian neocortex carries out the first stage of sensory information processing. Numerous studies have systematically investigated how sensory cortical neurons encode sensory information by means of selectively tuned responses to the relevant elementary features, such as the direction/speed/size of visual objects in the primary visual cortex (V1)[1–3], the acoustic frequency/loudness of sounds in the primary auditory cortex (A1)[4–7], and the identity/angle/torque of whiskers in the primary somatosensory cortex (S1)[8].

At the level of topographic maps of sensory features, primary sensory cortices in adult animals are well known to undergo significant shifts and refinements following sensory deprivation-recovery experiences[9], behavioural task training[10] or naturalistic experiences[11]. For example, in A1, both classical and operant conditioning resulted in a global shift of the tonotopic map towards the frequency of the conditioned tone[12]. However, although training-induced global shifts in topographical sensory feature maps have been extensively studied, how the firing properties of individual cortical neurons are transformed by training remains unknown, largely due to technical limitations.

There are two well-established techniques for acquiring sensory feature maps in the mammalian cortex. The first involves electrical recordings of neuronal firing at multiple arbitrary sites spread over the relevant sensory cortical region[13]. The second involves imaging with intrinsic optical signals[14] or bulk-loaded voltage-sensitive dyes[15] in a large frame, thereby sacrificing single-cell resolution to monitor the entire relevant sensory cortical region. In recent years, significant advances have been made in each approach. In the first approach (electrophysiology), modern high-density electrode arrays[16] have become available, enabling simultaneous recordings of hundreds to thousands of cells at high temporal resolution in actively behaving animal[17]. However, this approach still has methodological limitations, i.e., only active cells can be sampled, the gaps between sampled cells are arbitrary, and it is difficult to record the same cells over multiple days. In the second approach (imaging), with the development of chronic two-photon $Ca^{2+}$ imaging of neuronal populations with genetically encoded $Ca^{2+}$ indicators[18,19], a cell-by-cell analysis of neuronal responses can be performed over many days. For example, in monocular deprivation experiments, the ipsilateral visual stimulation–evoked $Ca^{2+}$ response amplitudes of some neurons in the binocular zone of V1 were significantly enhanced immediately after the deprivation period and then, after the recovery period, returned to the same level as before deprivation[20]. However, $Ca^{2+}$ imaging lacks the temporal resolution needed to identify spike firings over a large dynamic range[21], while voltage-sensitive imaging (whose latest significant advances achieve single-cell resolution in awake, behaving animals[22]) is limited by the viability of the recording preparation.

At present, the classical single-cell loose-patch recording technique[23] is still indispensable for precisely and reliably resolving spike firings without intracellular perturbation of a neuron. To study the spike firing properties of individual cortical neurons in animals following training, we used a combination of two-photon $Ca^{2+}$ imaging and single-cell loose-patch recording in awake behaving mice[19,24–26]. We performed targeted loose-patch recording of single neurons via online $Ca^{2+}$ signal analysis and patch pipette navigation under two-photon imaging guidance[27] and performed loose-patch recording simultaneously with live two-photon $Ca^{2+}$ imaging. With the help of this combined technique in layer 2/3 (L2/3) of A1 in animals following auditory associative training, we reveal a unique class of neurons, each of which exhibits high-rate bursting responses exclusively to the learned complex sounds but not to any of their constituent pure tones.

## Results

**An auditory associative training paradigm**. We trained head-fixed mice in a dark environment. Water was pumped at a constant latency (100 ms) after a brief (50 ms long) sound stimulus (Fig. 1a and Supplementary Movie 1; see Supplementary Methods for details). The pumping (20 ms) formed a water droplet on a spout. The water droplet remained at the spout until being consumed by the animal's voluntary licking or being replaced by a new droplet in the next trial. There was no other sensory stimulus, punishment or behaviour enforcing event. Each sound stimulus was followed by water pumping, regardless of the animal's behaviour.

Initially, we used broadband noise (BBN) with a constant waveform for both training and testing. A group of six animals started with a very low sound-evoked licking probability (Fig. 1b, session #1, $5 \pm 3\%$, mean $\pm$ s.e.m., same notation for all behavioural data in this section), which then reached $91 \pm 4\%$ in session #6 and then remained stably high in three more consecutive sessions (session #7, #8, #9: $96 \pm 4\%$, $94 \pm 2\%$, $93 \pm 5\%$) when the same sound-water relation was maintained on the rig. We then broke the sound-water relation by delivering sound stimulus without water on the rig. After 3 such detraining sessions, the sound-evoked licking probability dropped to $8 \pm 5\%$ (Fig. 1b, session #13, detrained, $P = 0.0095$, two-sided Wilcoxon signed-rank test). A control group of four animals (Fig. 1b, control-trained) first underwent the standard training sessions (#1 to #6) and then were kept in their home cage for 3 days. These control-trained animals at the testing timepoint showed a sound-evoked licking probability ($80 \pm 8\%$) that was slightly but not significantly ($P = 0.24$, two-sided Wilcoxon rank-sum test) lower than that in the last training session ($93 \pm 5\%$ at session #9). These data suggest that the behavioural effect of training was largely maintained over days without further training and could be reverted by detraining.

Next, we studied whether the licking behaviour was specific to the trained sound. We analysed the probability of spontaneously initiating a licking bout in each 1000 ms bin in the inter-trial intervals (continuous ongoing licking actions immediately after the sound-evoked licking were not considered spontaneous licking). The spontaneous licking probability was low and uniform at both naive and trained stages (Fig. 1c, naive: $3 \pm 2\%$, trained: $3 \pm 3\%$, $n = 6$ mice, $P = 0.75$, two-sided Wilcoxon signed-rank test); thus, training did not change the spontaneous licking probability. To test the specificity of behavioural response, a new group of 6 animals were trained with BBN in the same way as above and tested with the BBN as well as a list of pure tones. The results (Fig. 1d) showed that each tone-evoked licking probability (mean value in the range of 12–15% for each tone) was much lower than that evoked by BBN ($96 \pm 4\%$, $n = 6$ mice, **$P < 0.01$, two-sided paired bootstrap tests comparing each tone with the BBN). These results suggest that the licking behaviour in trained animals involved specific perception of the trained sound.

Next, we investigated whether animals could learn two different sounds. We modified the training rig with two spouts, where a 2 kHz tone was followed by water from the right spout, and a 12.1 kHz tone was followed by water from the left spout (Fig. 1e; for an example behaviour test, see Supplementary Movie 2). Note that the animals could be voluntarily probing on both spouts one after another. Thus, we considered only the first lick contact (within 1000 ms after sound stimulus onset) when calculating the licking probability (for the correct spout) evoked by each of the trained tones (Fig. 1f). In this set of experiments involving a new group of six mice, we also performed a psychometric test by using a range of pure tones that were not introduced in training. As shown in Fig. 1g, any non-trained tone, even including those only half an octave apart from a trained

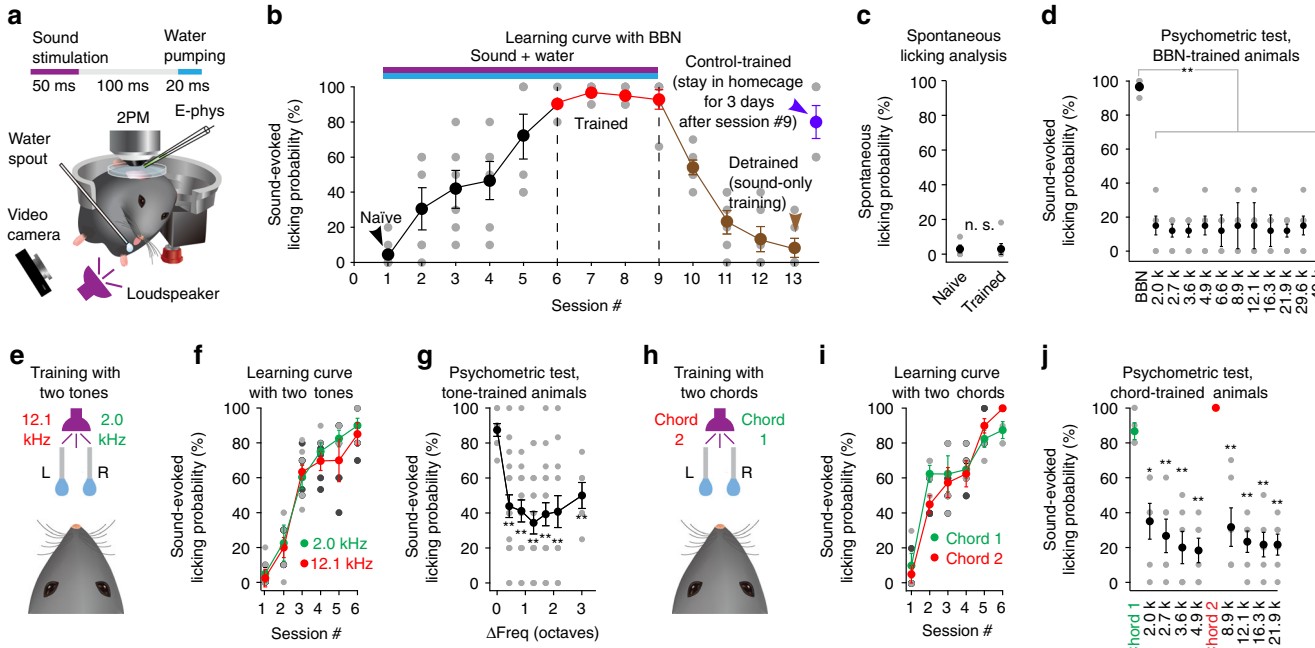

**Fig. 1 An auditory association training paradigm. a** Cartoon illustration of the sound-water association training paradigm and experiment setup. 2PM: two-photon microscope; E-phys: single-cell loose-patch pipette. **b** Learning curve, animals trained and tested with broadband noise (BBN), $n = 6$ mice. Error bars: $+/-$ s.e.m., same for all the other panels in Fig. 1. **c** Spontaneous licking analysis. n.s., $P > 0.05$. $n = 6$ mice. **d** Psychometric test in animals trained with BBN and tested with a range of pure tones; $P = 3.30e-3$, $P = 3.30e-3$, $P = 3.30e-3$, $P = 5.50e-3$, $P = 1.10e-3$, $P = 3.30e-3$, $P = 2.20e-3$, $P = 3.30e-3$, $P = 5.50e-3$, $P = 3.30e-3$, $P = 4.40e-3$, BBN versus different pure tones, respectively, two-sided paired bootstrap test, Bonferroni corrected, $n = 6$ mice. **e** Cartoon illustration of the 2-tone training paradigm. **f** Learning curve, animals trained with 2 tones (2.0 kHz or 12.1 kHz), $n = 6$ mice. **g** Psychometric test, 2-tone-trained animals, $\Delta$Freq (octaves): frequency difference between the test tone and one of the two trained tones (to the closer one); $P = 4.20e-3$, $P = 3.00e-3$, $P = 6.00e-4$, $P = 3.00e-3$, $P = 7.80e-3$, $P = 9.00e-3$, two-sided unpaired bootstrap test, the test tone versus other tones, respectively, Bonferroni corrected, $n = 6$ mice. **h** Cartoon illustration of the 2-chord training paradigm. **i** Learning curve, animals trained with 2 chords; $n = 6$ mice. **j** Psychometric test, 2-tone-trained animals tested with the two trained chords and each of their constituent tones; $P = 0.0128$, $P = 6.00e-3$, $P = 5.20e-3$, $P = 3.20e-3$, chord 1 versus its constituent tones, respectively, two-sided paired bootstrap test, Bonferroni corrected, $n = 6$ mice; $P = 5.60e-3$, $P = 1.20e-3$, $P = 1.20e-3$, $P = 8.00e-4$, chord 2 versus its constituent tones, respectively, two-sided paired bootstrap test, Bonferroni corrected, $n = 6$ mice. In psychometric testing sessions, different sounds were played in randomized trial orders, and behavioural responses until a total consumption of 20 droplets in one session were used for analysis. Data with error bars are presented as the mean ± s.e.m. **$P < 0.01$, *$P < 0.05$.

tone, evoked a licking probability (regardless of which spout) much lower than the licking probability (with correct choice of spout) evoked by a trained tone (non-trained tones: mean value in the range of 34%–50%; trained tones: 2 kHz: 90 ± 4%; 12.1 kHz: 85 ± 6%, $n = 6$ mice, **$P < 0.01$, two-sided unpaired bootstrap tests comparing each pair of non-trained tones with trained tones). This result suggests that animals could be trained to specifically distinguish two different tones against each other as well as against non-trained tones.

Cognitively meaningful sound stimuli in daily life for humans or animals alike are usually complex sounds. We next modified the 2-tone training protocol by using two synthesized chords (Fig. 1h). Chord 1 was composed of 2.0, 2.7, 3.6, and 4.9 kHz tones and was followed by water from the right spout; chord 2 was composed of 8.9, 12.1, 16.3, and 21.9 kHz tones and was followed by water from the left spout. Throughout all the training sessions, a new group of six mice exhibited a licking probability higher than 80% for each of the two chords (Fig. 1i). We also performed a psychometric test by using each of the tones on the composition list. The result (Fig. 1j, see also Supplementary Movie 3) showed a contrast in which, while the licking probability evoked by each chord was high (chord 1: 88 ± 5%; chord 2: 100 ± 0%, $n = 6$ mice), the licking probability evoked by any of the constituent tones was significantly lower (mean value in the range of 18–35%, $n = 6$ mice, **$P < 0.01$ for all two-sided paired bootstrap tests comparing each tone with each chord). Thus, at

the behavioural level, the chord-trained animals recognized each trained chord as a whole rather than as constituent features. This result is consistent with the above-shown result that BBN-trained animals exhibited very low licking probabilities evoked by pure tones (Fig. 1c) because the BBN can be regarded as a special chord that contains many tones of a broad range of frequencies. These results together suggest that the animals, as shown by their voluntary behaviour on the training rig, could indeed specifically distinguish trained stimuli against each other and against non-trained stimuli.

**Training induced an inhomogeneous transformation of neuronal population responsiveness.** It has been known that training experiences could enhance the overall neuronal population responsiveness in the relevant sensory cortex[12]. Thus, we next performed a set of two-photon Ca$^{2+}$ imaging experiments using a genetically encoded Ca$^{2+}$ indicator, GCaMP6f[19,27] (Supplementary Methods, Supplementary Movie 4), in A1 L2/3 to analyse single-cell responsiveness throughout our auditory training. By using widefield fluorescence imaging (Fig. 2a, b)[7] as well as retrograde labelling (Supplementary Fig. 1)[26,28], we established a reference cortical map of A1 for guiding cranial window surgery in the following two-photon imaging experiments. We then used AAV-GCaMP6f (e.g., Fig. 2c) in a new group of 10 mice that underwent both training and detraining with BBN.

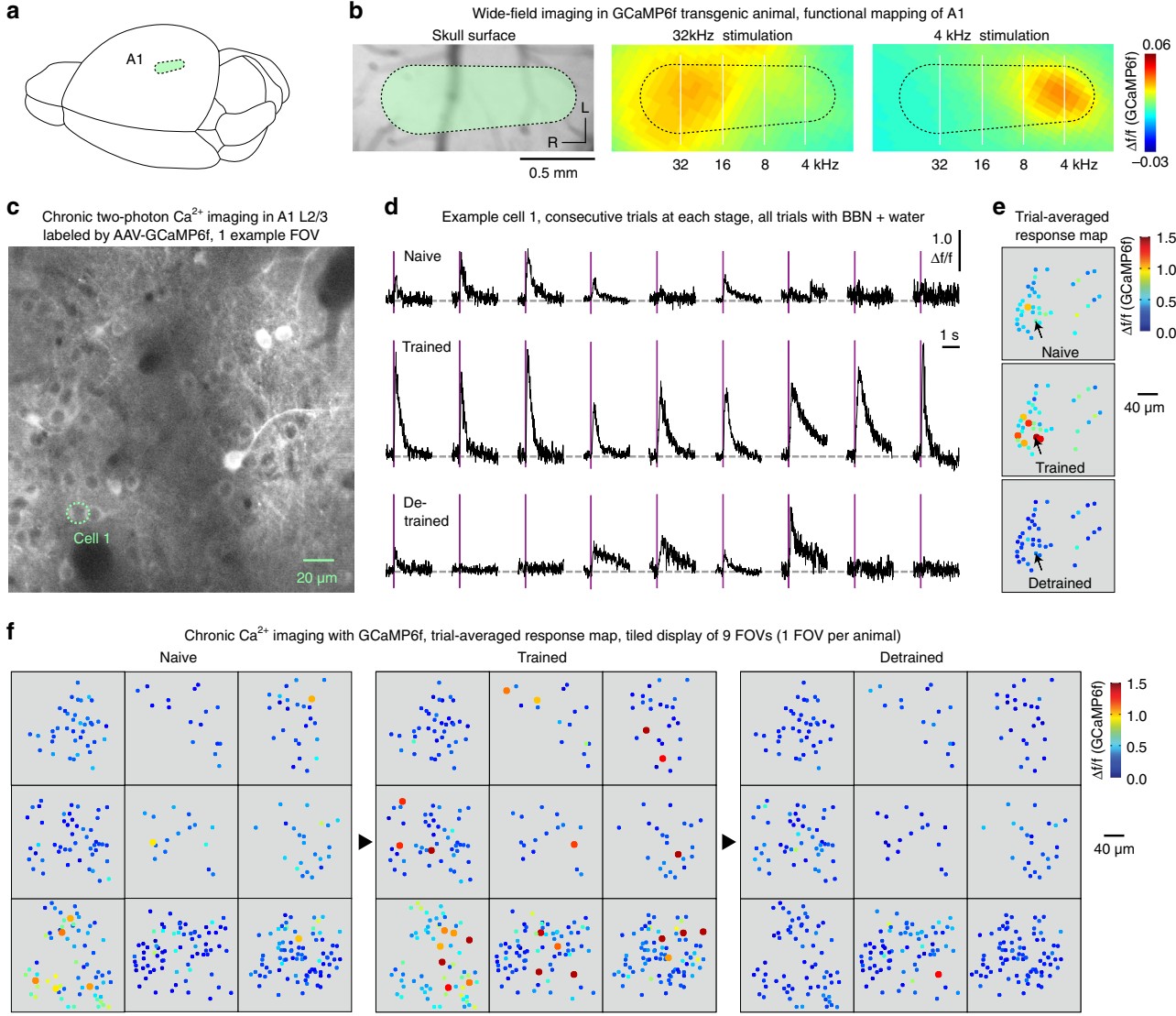

**Fig. 2 Chronic two-photon Ca²⁺ imaging with GCaMP6f in A1 L2/3 across training stages. a** Cartoon illustration of a mouse brain; the green patch outlines the primary auditory cortex (A1). **b** Left image: skull surface above the right auditory cortex, green patch outlines A1; middle and right images: widefield imaging (in a GCaMP6f-expressing mouse), fluorescence response to pure-tone stimulation with 32 and 4 kHz, respectively. Black dotted lines outline the A1 boundary. **c** Example two-photon image (averaged 100 frames) in A1 L2/3 of a head-fixed behaving mouse. **d** Consecutive trials of sound-evoked Ca²⁺ signals in an example cell (Cell 1 outlined by a dotted circle in (**c**)) at different training stages. **e** A pseudo-coloured map of sound-evoked Ca²⁺ response amplitudes (trial-averaged) at different stages (of the same field of view (FOV) as in (**c**), with an arrow pointing to the same example cell as in (**d**)). Cells with $\Delta f/f$ (GCaMP6f) $\geq 1.0$ are highlighted by enlarged dots (2-fold diameter). **f** Similar to panel **e**, pseudo-coloured neuronal response maps (minimum 3 trials per stage) for all FOVs (1 FOV per animal, 9 animals) at each stage.

Figure 2d shows one example cell that exhibited high-amplitude Ca²⁺ responses ($\Delta f/f$ (GCaMP6f) $\geq 1.0$) to the BBN stimulus in nine consecutive trials at the trained stage but much weaker and less reliable responses in the naive and detrained stages. However, the majority of cells in the same field of view (FOV) showed much weaker responses, even at the trained stage (Fig. 2e). To understand the spike firing properties underlying cellular Ca²⁺ response patterns, we performed a parallel set of calibration experiments with combined single-cell loose-patch recording and two-photon Ca²⁺ imaging in vivo under the same GCaMP6f expression conditions (Supplementary Fig. 2). Ca²⁺ transients of amplitude $\Delta f/f$ (GCaMP6f) $\geq 1.0$ (orange-red colours on the response map) corresponded to firing events consisting of more than 3 spikes per 150 ms time window. The complete dataset of chronic Ca²⁺ imaging (including the remaining nine animals, one imaging FOV performed in each animal, Fig. 2f)

showed that such high Ca²⁺-responsive cells (trial-averaged $\Delta f/f$ (GCaMP6f) $\geq 1.0$) at the trained stage were found in 9 out of the 10 FOVs. Altogether, 30 cells satisfied the criterion of trial-averaged $\Delta f/f$ (GCaMP6f) $\geq 1.0$ at the trained stage, out of a total of 423 cells pooled from the 10 animals. The Ca²⁺ response amplitudes (trial-averaged) of these 30 cells increased severalfold after training and decreased severalfold after detraining ($\Delta f/f$ values, naive: 0.48/0.31–0.54; trained: 1.4/1.2–1.8, detrained: 0.32/0.25–0.41; median/1st–3rd quartile, same notation for all subsequent data if not stated otherwise; $n = 30$ cells, two-sided Wilcoxon signed-rank test, ***$P < 0.001$ for both tests of naive versus trained and trained versus detrained). To exclude the possibility that the observed training-induced increase in response amplitude in this sparse subpopulation of cells (30/423, 7.1%) was due to an effect of thresholding on random fluctuations, we analysed single-trial responses in each of these 30

cells at the naive stage and at the trained stage. For each cell, the increment of the $\Delta f/f$ value (from the naive stage to the trained stage) was significant (two-sided Wilcoxon signed-rank test, $*P < 0.05$).

By comparison, when viewing all 423 imaged cells (including those 30 highly responsive cells) together as one population, we observed a much smaller although significant change in $Ca^{2+}$ response amplitudes (trial-averaged) following training and detraining ($\Delta f/f$ values, naive: 0.56/0.39–0.80; trained: 0.59/0.43–0.89; detrained: 0.47/0.38–0.58, $n = 423$ cells, two-sided Wilcoxon signed-rank test, $***P < 0.001$ for both tests). This statistical result, together with a complete cell-by-cell analysis (Supplementary Fig. 3), showed a strikingly inhomogeneous single-cell transformation following training: amidst a minor upward shift (~10%) in the overall neuronal population $Ca^{2+}$ responsiveness (to the trained sound, BBN), a drastic amplification (~3-fold) of $Ca^{2+}$ responsiveness occurred in sparse cells (~7.1% of population) in A1 L2/3.

**High-rate bursting: the firing properties of high-$Ca^{2+}$-responsive cells in trained animals.** Concerns on toxicity[29,30] and cell-by-cell variation in $Ca^{2+}$ sensing[31] caused by GCaMP6f expression may have undermined the finding of sparse highly $Ca^{2+}$-responsive cells in trained animals. Thus, we performed a new set of experiments by applying single-cell loose-patch recordings[23,32,33] in highly $Ca^{2+}$-responsive cells identified with a chemical $Ca^{2+}$ dye, Cal-520[24,27,34] (for example, see Supplementary Movie 5), with lesser concerns for toxicity and inhomogeneity in cell-by-cell $Ca^{2+}$ sensing. A detailed comparison between $Ca^{2+}$ imaging data obtained by the two different $Ca^{2+}$ indicators (Cal-520 and GCaMP6f) will be further demonstrated in the next section.

An example imaging FOV is shown in Fig. 3a, in which a cell with high $Ca^{2+}$ responses ($\Delta f/f$ (Cal-520) $\geq$ 1.5, the rationale of this criterion will be demonstrated in the next section) was targeted by a loose-patch pipette. Four consecutive trials of combined two-photon $Ca^{2+}$ imaging and loose-patch recordings (Fig. 3b, with magnified view of loose-patch recordings in Fig. 3c) showed that in each trial, a high-amplitude $Ca^{2+}$ transient occurred simultaneously with multiple spike firings in a time window of ~150 ms. In each of the seven trained mice in this set of experiments, we recorded at least one such highly responsive cell (there were multiple cells of high $Ca^{2+}$ responses in each animal, but we targeted no more than three cells per animal to minimize tissue damage), as shown in an overlaid display (nine cells, three trials shown for each cell) in Fig. 3d. Within an individual response event, spike waveform amplitudes tended to decay over time (fitted curve in Fig. 3d). All $n = 239$ recorded stimulation trials pooled from the same nine cells are shown as raster plots in Fig. 3e, for which the inter-spike interval (ISI) distribution histogram (Fig. 3f) had a prominent peak at approximately 10 ms (11/8.5–21 ms), corresponding to a high instantaneous firing rate of approximately 100 Hz (90/50–120 Hz). These two properties (~100 Hz instantaneous firing rate, decay in spike amplitude within event) were consistent with cortical neuronal burst firing as defined in previous studies[21,35,36]. Thus, we defined bursting spike responses as sound-triggered response events consisting of three or more spike firings in a 150-ms time window from the stimulus onset and with the presence of spike waveform amplitude decaying in the same time window. This definition also applies to other loose-patch recording data throughout this study.

While the last spike of the burst event overlapped with the onset of licking (Fig. 3g, 184/143–265 ms versus 250/120–370 ms, $P > 0.05$, two-sided Wilcoxon rank-sum test), the first spike of the

burst clearly occurred earlier than the onset of licking (29/22–49 ms versus 250/120–370 ms, $***P < 0.001$, two-sided Wilcoxon rank-sum test, $n = 239$ trials pooled for nine cells). Thus, these bursting responses were indeed evoked by the sound stimulus but not by tongue movement or water intake. We reconstructed the morphology of five recorded bursting responsive cells (from the two-photon z-stack images, for example see Fig. 3h), each showing a similarly rich dendritic tree and an axon extending out of the L2/3 of cortex, resembling stereotypical L2/3 cortical pyramidal neurons[37,38]. These data together reveal the existence of single neurons exhibiting trial-by-trial reliable burst firing responses to the trained sound (BBN) in A1 L2/3.

**Validating the emergence of high responsiveness (bursting) in sparse cells following training.** We established a new calibration graph of the $Ca^{2+}$ signal for the Cal-520 dye (Fig. 3i), including the nine highly responsive cells from above (Fig. 3e) and 7 other cells with different $Ca^{2+}$ response amplitudes that were randomly targeted, to provide a large sample of response events that covered a broad range of spikes per event (1–27 spikes; for example, see Supplementary Fig. 4). The burst detection threshold of $\Delta f/f$ (Cal-520) = 1.5 faithfully rejected singlet firing events (1–2 spikes, 0% false positive rate, $n = 49$ events) and was consistent with the previous threshold of $\Delta f/f$ (GCaMP6f) = 1.0 in the GCaMP6f imaging data (Supplementary Fig. 2); thus, both thresholds were conservative for detecting bursts (laying between 3 and 4 spikes). Therefore, in all subsequent sets of $Ca^{2+}$ imaging experiments without loose-patch recording, we applied a criterion ($\Delta f/f$ (Cal-520) $\geq$ 1.5 or $\Delta f/f$ (GCaMP6f) $\geq$ 1.0) on single-trial $Ca^{2+}$ signals to identify bursting response events in single trials, as well as on trial-averaged $Ca^{2+}$ signals per cell to identify bursting responsive cells from entire FOVs.

In all $Ca^{2+}$ imaging experiments, we used a high magnification objective (Nikon 40X/0.8) with a long working distance (3.5 mm) that was suitable for operating a patch pipette during live two-photon imaging in vivo, and we used a relatively high scanning zoom (2–4×) in the single-plane configuration to optimize morphological visibility and tissue stability for each cell in the FOV. These imaging quality factors were the only criteria for configuring the imaging FOV in each animal. Therefore, the number of imaged cells per FOV (and per animal) was not as high as that in other recent studies[39,40] by configuring the same type of two-photon microscope for large-FOV and volumetric imaging. Thus, we tested a large number of animals (8-naive and 18-trained animals with Cal-520 in a new set of experiments; 10 animals with GCaMP6f as the same dataset in Fig. 2) to compare the $Ca^{2+}$ imaging data with Cal-520 and with GCaMP6f and to verify the emergence of high $Ca^{2+}$ responsiveness (bursting) in sparse A1 L2/3 cells following training.

We established distribution histograms of single-cell responsiveness (trial-averaged $Ca^{2+}$ response $\Delta f/f$ values of each cell) for pooled neuronal populations in naive animals or trained animals. Interestingly, both the naive and trained distributions in both the Cal-520 and GCaMP6 datasets were highly skewed and could be well fitted with lognormal functions[41], appearing as symmetric Gaussian shapes when drawn with a logarithmic $x$-axis, as shown in Fig. 4. We found that the trained distribution had a significantly larger mean (on a log scale) than the naive distribution for both the Cal-520 and GCaMP6f datasets (both $***P < 0.001$, two-sided Wilcoxon rank-sum test). We then applied the above-established criterion to identify bursting responsive cells ($\Delta f/f$ (Cal-520) $\geq$ 1.5 or $\Delta f/f$ (GCaMP6f) $\geq$ 1.0). For all the imaged populations, we counted the number of bursting responsive cells. Bursting responsive cells were nearly absent in the eight-naive animals with Cal-520 imaging (0.2%,

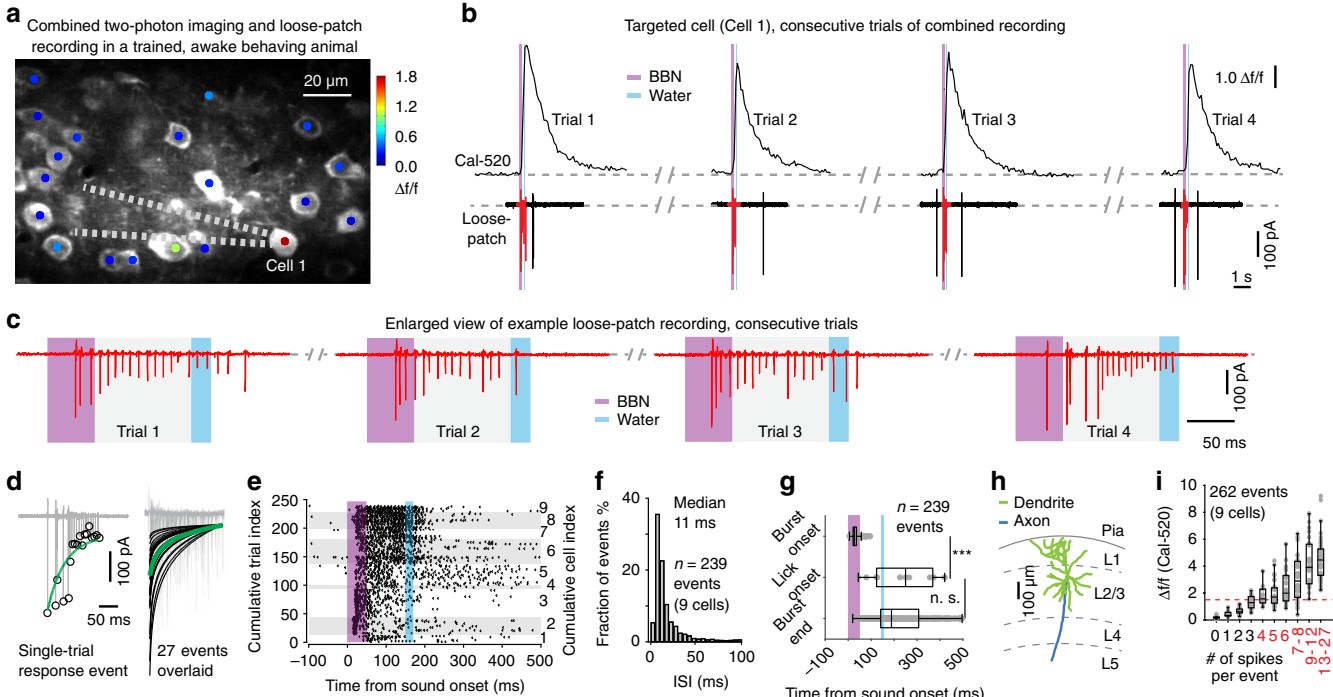

**Fig. 3 Two-photon Ca$^{2+}$ imaging with Cal-520 and single-cell electrophysiology in trained, awake behaving animals. a** Example of combined recording in a trained, behaving mouse. Light grey dotted lines illustrate the position of a micropipette (for loose-patch recording) targeting the cell marked by a red dot (pseudo-coloured response amplitude by two-photon Ca$^{2+}$ imaging with Cal-520). **b** Combined recordings on an example cell (Cell 1 in (**a**)); upper traces (Cal-520 imaging) and lower traces (loose-patch, recorded in voltage-clamp mode) are displayed on the same temporal scale. **c** An enlarged view of the loose-patch recording with ~200 ms of data around the stimulation timepoints (same data as in (**b**)). **d** One example response event (left subpanel) and 27 overlaid example events (three consecutive events per cell, nine cells, right subpanel) with exponential fitting of data points of spike peaks (appearing as troughs in the recorded traces). Black lines: fitting to individual events, green line: average of all fittings. **e** Spike raster plot of all trials for 9 bursting responsive cells (cells sorted by their average response strength). **f** Inter-spike interval (ISI) histogram for all the response events as in (**e**). **g** Box plots show the timing of burst onset, lick onset and burst end. $P = 1.28e{-}7$ (Lick onset versus Burst onset), $P = 0.98$ (Lick onset versus Burst end), two-sided Wilcoxon rank-sum test, Bonferroni corrected, Burst $n = 239$ events, Lick $n = 11$ events, ***$P < 0.001$, n.s., $P > 0.05$. **h** Side-view projection of the 3D reconstruction of an example bursting responsive cell. **i** Box plots showing relations between single-trial Ca$^{2+}$ response amplitude and the number of spikes per response event, $n = 21, 20, 29, 25, 10, 29, 26, 26, 28, 48$ events, respectively, involving response events pooled from the nine cells shown in (**e–g**) and another seven non-bursting cells. The red dashed line indicates $\Delta f/f = 1.5$. In this figure, violet shade: sound stimulation time, light blue shade: water pumping time. Boxes represent 25th and 75th percentiles (Q1 and Q3), i.e., interquartile range (IQR), central bars indicate the median, and whiskers indicate Q1-1.5 × IQR and Q3 + 1.5 × IQR.

2/1314 cells; Fig. 4a left). In contrast, in the 18-trained animals with Cal-520 imaging, the bursting responsive cells constituted 4.7% of the population and were statistically significant (52/1112 cells; Fig. 4a right, ***$P < 0.001$, Fisher's exact test). Importantly, at least one bursting responsive cell was found in each of the 18-trained animals (imaged with Cal-520). The Cal-520 imaging results (Fig. 4a) largely reproduced the GCaMP6f imaging results (Fig. 4b, same dataset as in Fig. 2), where 2.1% (9/423, Fig. 4b left) of cells at the naive stage and a significantly increased fraction of 7.1% (30/423, Fig. 4b right, ***$P < 0.001$, Fisher's exact test) of cells at the trained stage were bursting responsive, and in 9 of 10 FOVs, we observed at least one training-transformed bursting responsive cell (see also Fig. 2f). There was a minor, statistically nonsignificant difference in the bursting cell ratio (GCaMP6f: 7.1%, $n = 10$ mice; Cal-520: 4.7%, $n = 18$ mice; $P = 0.056$, Chi-square test). The above analysis for comparing the Cal-520 and GCaMP6f datasets showed the same significant results: training induced an upward shift of overall population responsiveness and the emergence of high responsiveness (bursting) in sparse A1 L2/3 cells in trained animals. Moreover, the cell-by-cell analysis of the GCaMP6f dataset (Fig. 2 and Supplementary Fig. 3) showed that the bursting responsiveness at the trained stage emerged primarily by the strong amplification (~3-fold) of response amplitudes in a few weakly responsive cells but not by up-shifting

the response amplitudes of those few already highly responsive cells at the naive stage.

**Reliable sound-evoked bursting responses regardless of behavioural motivation.** Previous literature[42,43] has suggested that cortical neuronal responses could be modulated by behavioural motivational drive. Thus, we analysed the sound-evoked bursting responses and licking actions on a trial-by-trial basis in trained animals. In some imaging (with Cal-520) sessions, we applied many stimulation trials of the same trained sound (BBN) followed by water delivery to perform recordings of sound-evoked neuronal responses sequentially at different imaging FOVs (Fig. 5a–d, in 7 out of 18 BBN-trained animals we imaged at multiple FOVs). After ~20–30 trials in an imaging session, sound-evoked licking probability became significantly lower, most likely because the animal no longer became thirsty (Supplementary Methods). Nevertheless, bursting responses ($\Delta f/f \geq 1.5$ (Cal-520) in single trials) remained highly reliable (e.g., example cell #6 in Fig. 5d) even when the trained animal did not lick at all.

We binned each 10 consecutive trials as subsessions to calculate the sound-evoked bursting response probability and licking probability for each animal (Fig. 5e). The sound-evoked licking probability started at a high level (≥80%) but gradually decreased

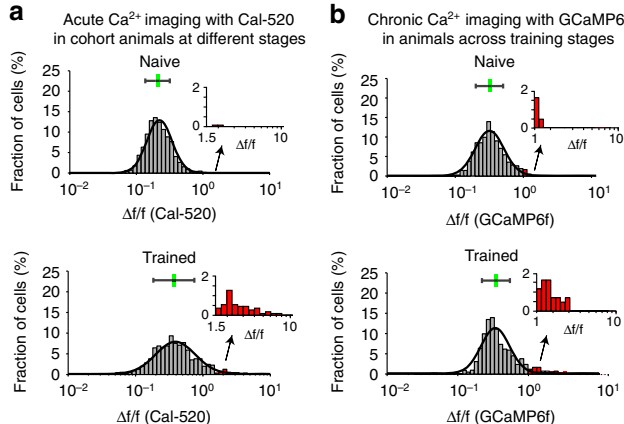

**Fig. 4 Comparing Ca$^{2+}$ imaging datasets using Cal-520 with those using GCaMP6f.** Histograms of trial-averaged Ca$^{2+}$ response amplitudes for pooled populations of cells (**a** upper: $n = 1314$ cells pooled from eight naive animals, lower: $n = 1112$ cells pooled from 18 animals imaged with Cal-520; **b** upper and lower: the same $n = 423$ cells pooled from 10 animals imaged with GCaMP6f). Each histogram is fitted by a lognormal function, and the inset panel shows an enlarged view of the histogram tail for $\Delta f/f \geq 1.5$ (Cal-520) and $\Delta f/f \geq 1.0$ (GCaMP6f).

(to ≤20%, one-way ANOVA, $P = 5e{-}5$; $n = 8$ subsessions) in an animal-dependent manner. In contrast, the probability of sound-evoked bursting responses remained at a high level of ≥80% throughout all subsessions (one-way ANOVA, $P = 0.13$; $n = 8$ subsessions). These data reveal the existence of trial-by-trial reliable neuronal bursting responses in A1 L2/3 (evoked by the trained sound), which were independent of the changes in behavioural motivation level or behavioural outcome over repeated trials.

**Holistic bursting (HB) cells**. At the behavioural level, the animals trained with two chords (Fig. 1i) recognized each of the trained chords as a whole but not as constituent tones (Fig. 1j). We next investigated A1 L2/3 neuronal response properties (imaging with Cal-520) in a new group of 7 mice trained with the two chords and tested with each chord as well as each of their constituent tones. We applied the above-established $\Delta f/f$ (Cal-520) ≥ 1.5 criterion for trial-averaged signals to detect bursting responsive cells. Strikingly, in chord-trained animals (Fig. 6a), we found some cells in A1 L2/3 that exhibited bursting responses only to a specific trained chord but not to any of the individual constituent tones of either chord (see the first example cell in Fig. 6b; also see Supplementary Movie 6). Because the bursting activity of these neurons represents an integration of all components of the complex auditory stimulus, we referred to them as HB cells. Accordingly, since the BBN could also be regarded as a combination of multiple pure tones (see Fig. 1c), in another group of 6 BBN-trained animals (Fig. 6c) that were tested with the BBN as well as a range of tones, we also found a few cells that exhibited bursting responses to the BBN in a holistic manner (see the first example cell in Fig. 6d). In addition, for both groups of chord-trained and BBN-trained animals, we found a few quasi-holistic bursting (qHB) cells that exhibited bursting responses to a complex sound (a chord or BBN) as well as to one or several (but not all) of its component tones (see the second example cells in Fig. 6b and d). Furthermore, we also found analytic bursting (AB) cells that exhibited bursting response only to one or a few tested tones (see the third example cells in Fig. 6b and d).

Overall, bursting responsive cells (of all different properties: HB, qHB and AB) were rare, and the majority (>90%) of all the observed cells were non-bursting (NB) to any of the tested sound stimuli. The sound-evoked response properties of all the bursting cells are shown in Fig. 6e (2-chord-trained animals, 18 HB cells, 5 qHB cells and 19 AB cells out of a total of 570 cells in 19 imaging FOVs in seven animals) and Fig. 6f (BBN-trained animals, 22 HB cells, 8 qHB cells and 37 AB cells out of 681 neurons in 22 FOVs in 6 animals).

Note that the HB, qHB and AB cells could be identified only after testing the complex sound-trained animals with different pure tones. We found that the number of HB cells per trained complex sound was significantly higher than that of qHB cells or AB cells per tested pure tone. (Fig. 6e, 2-chord-trained animals tested with the 2 chords and 8 tones: HB versus qHB cells, $P = 0.012$; HB versus AB cells, $P = 0.0011$, Chi-square test; Fig. 6f, BBN-trained mice tested with the BBN and 11 tones, HB versus qHB cells, $P = 0.016$, HB versus AB cells, $P = 0.00041$, Chi-square test). Interestingly, for the HB cells in the 2-chord-trained mice, we found that the response ($\Delta f/f$ (Cal-520)) evoked by the preferred chord was significantly greater than either the summation of 4 tone-evoked responses, the best tone response, or the non-preferred chord response (Fig. 6g, preferred chord: 2.93/2.31–4.25, sum of four constituent tones: 1.49/1.30–1.95, $P = 0.003$, $z = 2.94$; best tone response: 0.61/0.48–0.96, non-preferred chord response: 0.41/0.23–0.64, $P < 0.001$; two-sided Wilcoxon signed-rank test; $n = 18$ cells). Therefore, these results suggest that the HB cells found in the trained animals exhibit not only a simple preference to a trained complex sound but also a highly nonlinear summation.

Taken together, these data reveal the existence, in A1 L2/3 of mice trained with complex sound(s), of distinct sparse sets of HB/qHB cells that possessed bursting responses to each trained complex sound (chord or BBN) as a whole, with the co-existence of AB cells that possessed enhanced tuning response properties to various individual tones.

**HB and qHB cells together carry perfect information of learned complex sounds**. Since individual neurons often show a large trial-by-trial response variability, much effort was made to show that a sufficiently large neuronal population could nevertheless carry information about complex objects reliably in single trials by means of various population coding schemes[44,45]. Here, in contrast, we found sparse sets of individual HB/qHB cells in trained animals that were highly reliable in their responses to the trained complex stimuli. What could be the contribution of such sparse cells to neuronal population information coding?

We used a linear support vector machine (SVM) algorithm to read out the classification information from a single-trial neuronal population response pattern (a vector consisting of $\Delta f/f$ values of neurons in a defined population), as illustrated in the cartoon in Fig. 7a. Here, the information of a trained sound is defined as the single-trial accuracy to discriminate that specific trained sound (labelled as the target class) from any non-trained or the other trained sound (together labelled as the nontarget class). We performed the SVM test in various scenarios (Fig. 7b–d, see Supplementary Methods for detailed procedure and results). We recruited different numbers of cells into a population in specific orders (either non-bursting cells first or bursting cells first) and ran the SVM on a single-trial population response vector to discriminate the sound stimulus. The results are expressed in curves showing the single-trial decoding accuracy ($A_c$) versus the number of cells recruited in the population ($N_p$). Importantly, to enable the analysis to report the contribution of individual cells, we also configured hypermouse (Fig. 7c, d) datasets by concatenating data from multiple FOVs from different animals under the same experimental conditions to

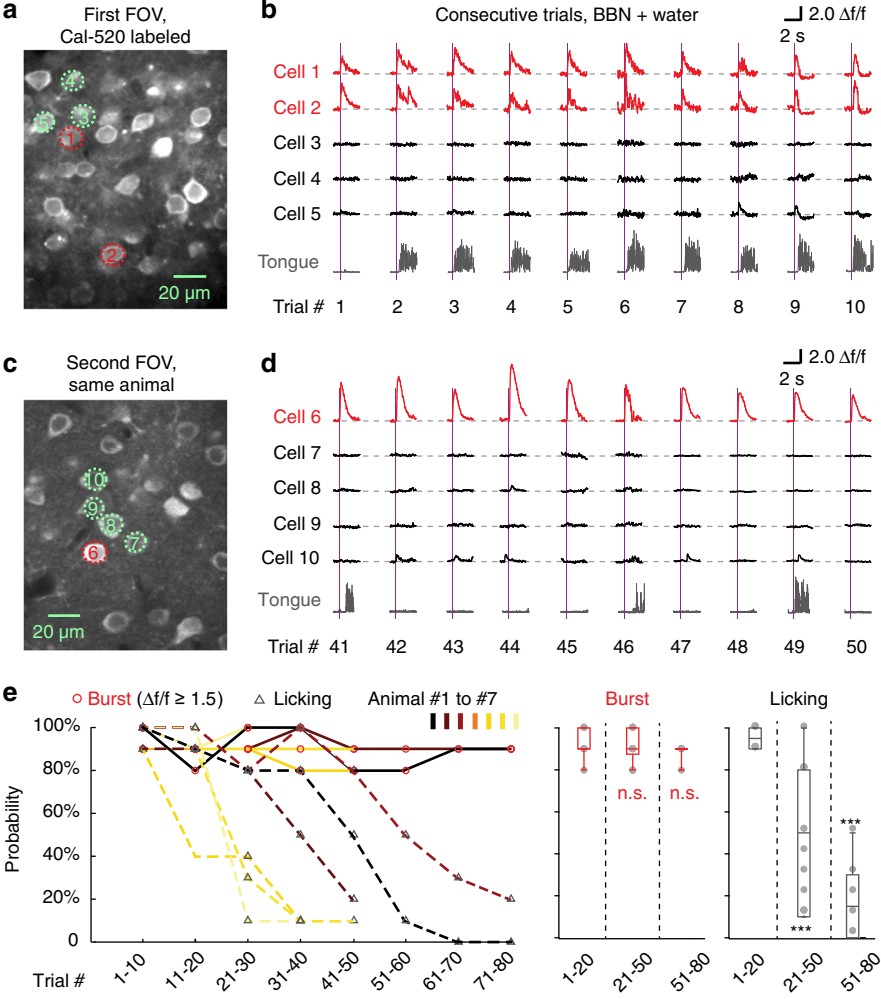

**Fig. 5 Trial-by-trial analysis of sound-evoked bursting response and licking behaviour. a** First imaging FOV in a head-fixed mouse in a testing session of sound-evoked licking behaviour. **b** Sound-evoked Ca²⁺ responses (upper) from the cells outlined with dashed circles in (**a**) and corresponding licking action traces (lower, grey traces). **c**, **d** The same arrangement as in **a** and **b**, obtained at the second imaging FOV during the same experiment session, with the cumulative trial index counting from 41. **e** Left graph, scatter plot shows the bursting response probabilities (red circles) and the sound-evoked licking rates (grey triangles) with respect to the trial index range. Datapoints are collected from $n = 7$ mice, and the licking probability data points are connected by coloured dashed lines to indicate each individual animal. Middle and right boxplots, summary of bursting response probabilities (red boxplots) and licking rates (grey boxplots) for the same datapoints on the left graph. Middle: $P = 1.00$ (1–20 versus 21–50), $P = 0.34$ (1–20 vs. 51–80), two-sided Wilcoxon rank-sum test, Bonferroni corrected, Burst $n = 14, 17, 6$ events, respectively; Right: $P = 2.44e{-}5$ (1–20 vs. 21–50), $P = 6.92e{-}4$ (1–20 vs. 51–80), two-sided Wilcoxon rank-sum test, Bonferroni corrected, Licking $n = 14, 17, 6$ events, respectively; n.s., $P > 0.05$, ***$P < 0.001$. Boxes represent Q1 and Q3, central bars indicate the median, and whiskers indicate Q1-1.5 × IQR and Q3 + 1.5 × IQR.

reduce the potential contribution of noise correlations in populations of neurons within the same animal.

The results showed that when only NB cells were recruited, the single-trial decoding performance ($A_c$) increased monotonically with the number of cells ($N_p$). The best performance achieved with the maximum available number of NB cells in each scenario is as follows (the chance level was 50% in all scenarios). Figure 7b, middle graph: $A_c = 84.4 \pm 0.5\%$ in one example mouse of the BBN-trained scenario; Fig. 7c, middle graph: $A_c = 94.6 \pm 0.5\%$ in the hypermouse of the BBN-trained scenario; Fig. 7d, middle graph: $A_c = 73 \pm 4\%$ in the hypermouse of the 2-chord-trained scenario. In contrast, the few HB and qHB cells together supported perfect classification ($A_c = 100\%$ in all scenarios). On the other hand, AB cells sometimes improved the classification and sometimes didn't, depending on whether HB and qHB cells were present or not (see Supplementary Methods for details of the analysis and result). These results show that linear decoding of a large set of NB cells could achieve a significant discrimination

accuracy of single trials, but the very few HB/qHB cells could achieve a perfect discrimination under the same conditions.

## Discussion
By simultaneously combining two-photon Ca²⁺ imaging of neuronal populations with targeted single-cell loose-patch recordings in awake behaving mice, we found that the sound-water association training-transformed distinct sparse subsets of A1 L2/3 neurons from a non-bursting mode to a bursting mode in response to a trained sound. The temporal relations of response events (Fig. 3) demonstrated that those A1 L2/3 bursting responses were not induced by licking or water intake. Furthermore, in trained animals, the licking probability evoked by the trained sound dropped over repeated trials of stimulation, yet the sound-evoked bursting responses in sparse A1 L2/3 cells continued to occur with a high reliability (Fig. 5). Thus, the bursting responses that emerged in sparse A1 L2/3 cells reliably

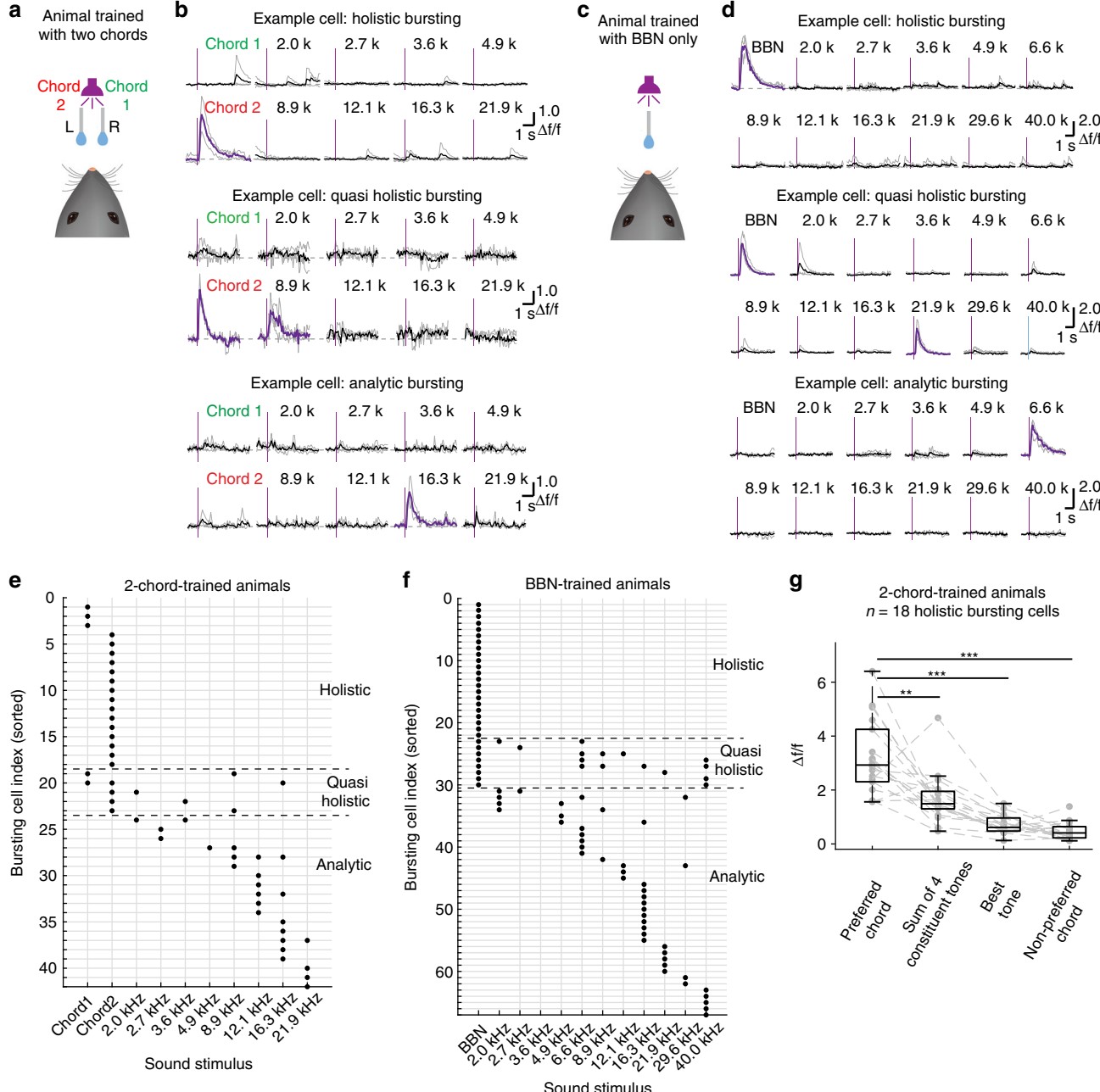

**Fig. 6 Holistic, quasi-holistic and analytic bursting cells in animals trained with complex sounds. a** Cartoon showing a 2-chord-trained animal. **b** Ca$^{2+}$ signals evoked by different sound stimuli using each chord and each pure tone for three example cells. Single trials (grey traces) and their averages (black and purple traces) are overlaid. Vertical bars indicate sound stimulus onset time. **c, d** Similar arrangement as in panels **a** and **b** showing different example cells from BBN-trained mice. **e** Summary for holistic, quasi-holistic and AB responsive cells identified out of 570 cells (pooled from all 19 imaging FOVs in 7 chord-trained animals tested with the two chords and 8 constituent tones); each dot represents the bursting response (trial-averaged Ca$^{2+}$ response amplitude, $\Delta f/f$ (Cal-520) $\geq$ 1.5) to the corresponding stimulus. **f** Same arrangement as panel **e**; the summary for bursting responsive cells out of a total of 681 cells (from all 22 imaging FOVs in 6 BBN-trained animals that were tested with BBN and pure-tone stimuli). **g** Comparing the response to the chord with the responses to constituent tones in the 2-chord-trained mice. Each data point represents the trial-averaged Ca$^{2+}$ response amplitude for a certain category. Sum of four constituent tones means summing the responses evoked by each of the four constituent tones that correspond to the preferred chord. Datapoints for each cell were individually calculated (i.e., the preferred chord could be a different one, or summing different tone responses), $P =$ 0.0099 (chord versus sum of 4 tones), $P = 5.88e{-}4$ (chord versus best tone), $P = 5.88e{-}4$ (chord versus non-preferred chord), two-sided Wilcoxon signed-rank test, Bonferroni corrected, $n = 18$ cells. ***$P < 0.001$, **$P < 0.01$. Boxes represent Q1 and Q3, central bars indicate the median, and whiskers indicate Q1-1.5 × IQR and Q3 + 1.5 × IQR.

represented the trained sound but not behavioural motivation or behavioural outcome.

In animals trained with complex sounds (BBN or chords) and tested with these sounds as well as a range of pure tones, we

discovered the co-existence of three distinct subsets of neurons in A1 L2/3: HB cells that exhibited bursting response exclusively to the learned complex sounds; qHB cells that exhibited bursting response to the learned complex sounds as well as to one or

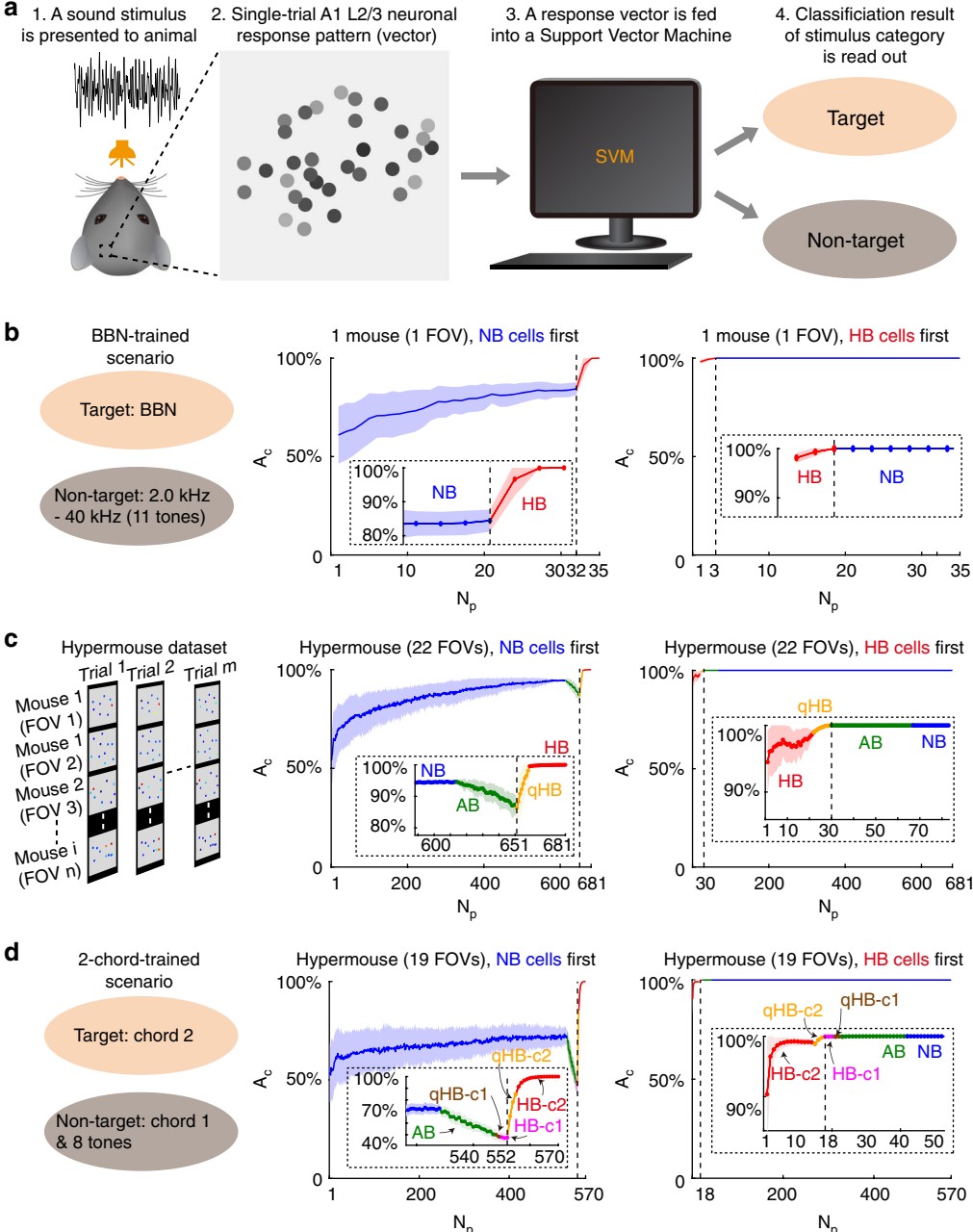

**Fig. 7 Holistic/quasi-holistic bursting cells carry perfect information of trained complex sounds. a** Cartoon illustration of the machine classification task. Note that in practice, the biological experiments and the machine classifications were individually and sequentially performed, but the logic of information flow was as shown in the model. **b** Left cartoon: illustration of a scenario of a task with BBN-trained mice. Middle and right graphs: accuracy of classification ($A_c$) as a function of the neuronal population size ($N_p$) for one example animal. Coloured lines and shades indicate the mean and variance in the $A_c$ value corresponding to each $N_p$ value. Inset panels outlined by dashed boxes show the magnified views of the $A_c$–$N_p$ relation curve near the transition point. **c** Same analysis as in (**b**) performed in a hypermouse (as illustrated in the left cartoon). A neuronal response pattern of a given trial is made by concatenating the patterns of all involved FOVs of the tested mice together; patterns in each FOV for the same sound with the same chronologically ordered trial index were put together. **d** Same analysis as in panel **c** but for a different hypermouse dataset from the chord-trained animals. Shading indicates s.d.

several of the constituent tones; and AB cells that exhibited bursting response to pure tones only. The co-formation of HB/qHB/AB cells in L2/3 of the sensory cortex can be explained in light of a series of in vivo subthreshold single-cell dendritic imaging studies[37,38,46]. This ubiquitous finding of the dendritic organization of sensory input features[47] showed that individual L2/3 primary sensory cortical (V1, A1 or S1) neurons, regardless of either their location on topographical maps of elementary

sensory features or their output preferences, received multiple synaptic inputs of highly diverse feature specificities throughout entire dendrites in a widely distributed manner. Accordingly, our results here show that a few L2/3 primary sensory cortical neurons exhibited a strong response to specific conjunctions of multiple elementary features as HB/qHB cells, while a few other L2/3 neurons exhibited an enhanced tuning response to individual single elementary features as AB cells. This reservoir of

single-cell multi-input operating logic found in awake behaving animals is consistent with those reported by previous in vitro studies[48,49], including a recent study on human L2/3 cortical neurons[50].

State-of-art neuromorphic computing hardware[51] typically supports a classical integrate-and-fire single-unit operating logic[52]. Here, our findings could inspire a new HB single-unit operating logic to be deployed for the purpose of storing and retrieving the trained complex information at a high accuracy and specificity (Fig. 7). Sparse subsets of standard integrate-and-fire model neurons can be transformed to deliver a high-rate, long-duration burst of output spikes (instead of 1 spike) upon specific conjunctions (but not summations) of multi-input activations. Beyond being an inspiration to brain-like computing and artificial intelligence technologies[53,54], burst firing is well known to have significantly more efficacy than singlet firing in establishing synaptic transmission and plasticity[55] in biological neurons. In particular, the bursts recorded in our study had an instantaneous firing rate of 50–120 Hz and a firing duration of 100–200 ms, matching well that typically applied in the stimulation protocols for optimally inducing synaptic long-term plasticity in both rodent[56] and human brain tissues[57]. Overall, our finding of the unique class of bursting neurons representing learned complex sounds in the auditory cortex demonstrates how single cortical neurons represent complex objects as a whole rather than the sum of their parts.

## Methods

**Animals**. C57BL/6J male mice (2–3-months old) were provided by the Laboratory Animal Center at the Third Military Medical University. The mice were housed in a temperature- and humidity-controlled room on a cycle of 12-h light/dark (lights off at 19:00). All experimental procedures were performed in accordance with institutional animal welfare guidelines with the approval of the Third Military Medical University Animal Care and Use Committee.

**Auditory stimulation**. Sound stimuli were delivered by an ED1 electrostatic speaker driver and a free-field ES1 speaker (both from Tucker Davis Technologies)[26,27,37]. The distance from the speaker to the mouse ear (contralateral to the imaged A1) was ~6 cm. The sound stimulus was produced by a custom-written, LabVIEW-based program (LabVIEW 2012, National Instruments) and transformed to analogue voltage through a PCI6731 card (National Instruments). All sound levels tested with a microphone placed ~6 cm away from the speaker were calibrated by a pre-polarized condenser microphone (377A01 microphone, PCB Piezotronics). All the data were obtained at 1 MHz via a data acquisition device (USB-6361, National Instruments) and examined by our custom-made LabVIEW program. For BBN (BBN, bandwidth 0–50 kHz), the sound level was ~65 dB sound pressure level (SPL). We generated a waveform segment of BBN and used the same waveform segment for all experiments involving BBN. For testing neuronal response characteristics, we used 11 pure tones, the frequencies of which were logarithmically spaced in the range of 2–40 kHz. The sound levels were ~74 dB SPL (2.0–10 kHz) and ~67 dB SPL (10–40 kHz), respectively. For the 2-chord training experiments, chord 1 consisted of 2.0, 2.7, 3.6, and 4.9 kHz, while chord 2 consisted of 8.9, 12.1, 16.3, and 21.9 kHz. The sound level of chord 1 was ~78 dB SPL, and the sound level of chord 2 was ~71 dB SPL. The background noise level was kept at ~55 dB SPL for all experiments. As described in our previous reports[26,58], low frequencies (<1 kHz) were major components of background noise. With a spectral density of ~33 dB/sqrt (Hz), the peak of background noise is below 1 kHz. Neither visible light nor other sensory stimuli were present. The duration of a sound stimulus (tone, chord or BBN) was 50 ms.

**Training, detraining and testing**. Before training, the animal was implanted with the headpost under isoflurane anaesthesia and then allowed to recover for 5 days. During the training period, the animal was head-fixed to the recording rig and received water exclusively on the training rig. A droplet of water was formed at a spout by automatically controlled pumping (pumping duration, 20 ms) at 100 ms after the end of the sound stimulus (in total, 50 + 100 = 150 ms from the stimulus onset) (Fig. 1a). Licking actions were monitored with a camera (frame rate 30 Hz) under infrared illumination that was invisible to the animal. Sound stimuli were delivered without any cues at random inter-trial intervals (in the range of 5–10 s, longer than the duration of a licking action). The rationale of using a randomized inter-trial interval setting was to avoid the possible effect of the rhythmic predicative responses that have been known to exist in mice[26].

Water droplets remained at the spout after being delivered so that the animal could always obtain water if ever it voluntarily made a licking action at any time

after water was delivered. If the animal had not licked before the next trial occurred, a new droplet would replace the previous one at the spout. There was no cue, stimulus or reward/punishment object beyond the sound and the water. The spout was positioned at a distance of approximately 3–4 mm from the animal mouth (and with no visible ambient light) such that the animal had to voluntarily stretch out its tongue to probe and acquire water droplets on the spout. A success sound-evoked licking event was defined as an event in which the animal initiated a licking action within 1000 ms from sound stimulus onset. One droplet had an ~5 µl volume, and 20 droplets of water together had a volume of ~0.1 ml. For one experimental session, only those trials until mice acquired 20 droplets of water were considered effective trials and used for calculating the sound-evoked licking probability (except for the data shown in Fig. 5, where the licking probability was calculated for each 10 consecutive trials, the result of which prompted us to define the 20-droplet criterion retrospectively).

Animals were considered trained if the sound-evoked licking probability (mean value) was ≥80%. We did not apply any punishment for incorrect licking timing. The training rigs inside isolation boxes and the testing rig under a two-photon microscope were nearly identical, and the animals did not show obvious signs of discomfort on either of the rigs. One experimental session typically involved 40–100 stimulus events. The total amount of water consumed by an animal on the rig was sufficient to prevent dehydration, which was confirmed by the body weight control (<20% weight loss throughout all the experimental days)[27].

There were three different training scenarios with different association sounds in this study applied to different groups of animals. The association sounds were code-named as follows: BBN: the BBN with one spout, 2-tone: two pure tones with two spouts (2.0 kHz with the right spout and 12.1 kHz with the left spout), and 2-chord: two chords with two spouts (chord 1 consisting of 2.0, 2.7, 3.6 and 4.9 kHz with the right spout; chord 2 consisting of 8.9, 12.1, 16.3 and 21.9 kHz with the left spout). All groups of animals were tested with the same sound used for training to assess the behaviour performance over sessions, i.e., the learning curve. In some of the BBN and 2-chord training groups of animals, a full range of different pure tones, including the training tones (kHz: 2.0, 2.7, 3.6, 4.9, 6.6, 8.9, 12.1, 16.3, 21.9, 29.6 and 40.0) and the BBN, were used to test neuronal responses. Water was always delivered on the corresponding spout (for testing the 2-chord-trained animals, delivering on the right spout for the 4 lower tones and on the left spout for the four higher tones).

In detraining sessions (Figs. 1 and 2), all configurations and parameters of the rig and session were exactly the same as those in training, except that there was no water delivery accompanying the sound stimulus. Note that the two-photon imaging testing for detrained animals was under the same condition as that in the testing session after training, i.e., there was water delivery following the sound stimulus.

**Two-photon Ca²⁺ imaging in A1 L2/3**. For two-photon imaging in head-fixed awake mice[27], we glued a titanium head post to the skull for head fixation under isoflurane anaesthesia. Three days after surgery, animals received 1 ml of water supply per day for 2–3 days and then underwent either training or testing sessions with water deprivation in their home cages (for details, see the section above).

For acute imaging experiments with Cal-520 AM, we exposed the right A1 of the mouse[4,37]. In brief, the animal was anaesthetized by isoflurane and kept on a warm plate (37.5 °C). The skin and muscles over the auditory cortex were removed after local lidocaine injection. A custom-made plastic chamber was glued to the skull with cyanoacrylate glue (UHU), followed by a small craniotomy (~2 mm × 2 mm) (the centre point: Bregma –3.0 mm, 4.5 mm lateral to midline). Afterwards, the animal was transferred to the recording rig. The craniotomy was filled with 1.5% low-melting-point agarose. The recording chamber was perfused with normal artificial cerebral spinal fluid (ACSF) containing 125 mM NaCl, 4.5 mM KCl, 26 mM NaHCO₃, 1.25 mM NaH₂PO₄, 2 mM CaCl₂, 1 mM MgCl₂ and 20 mM glucose (pH 7.4 when bubbled with 95% oxygen and 5% CO₂). Cal-520 AM was dissolved in DMSO with 20% Pluronic F-127 to a final concentration of 567 µM for bolus loading. Pressure (~600 mbar, 3 min) was applied to the glass to load dye solution. Ca²⁺ imaging was performed ~2 h after dye injection and lasted for up to 8 h[7,27,59].

For chronic imaging experiments with GCaMP6f, standard sterile protocols were used in our surgery[27]. Fourteen days after virus (AAV2/8-GCaMP6f, ~1 × 10¹³ vg/ml, Obio Technology Corp., Ltd, Shanghai) injection, we exposed the right A1 of the mouse as described above. A small plastic chamber was glued to the skull with cyanoacrylate glue (UHU). Then, a piece of bone was removed and replaced by a smaller coverslip (1.5 mm in diameter, below) and a larger coverslip (2.5 mm in diameter, above), which were sealed with ultraviolet cured optical adhesives (Norland Products Inc., USA) and dental acrylic. In the experiments for electrophysiological recordings, these coverslips were removed. Antibiotics (cefazolin, 500 mg/kg, intraperitoneal injection, North China Pharmaceutical Group Corporation) were administered for 3 days. The animal was housed for a week's recovery. Then, the mouse was engaged in training and detraining with protocols as described in the section above.

Two-photon imaging was performed with a custom-built two-photon microscope system based on a 12.0 kHz resonant scanner (model "LotosScan 1.0", Suzhou Institute of Biomedical Engineering and Technology)[38,60]. Two-photon excitation light was delivered by a mode-locked Ti:Sa laser (model "Mai-Tai

DeepSee", Spectra Physics) and a 40×/0.8 numerical aperture (NA) water-immersion objective (Nikon) was used for imaging. For $Ca^{2+}$ imaging experiments, the excitation wavelength was set to 920 nm. The typical size of the FOV was ~200 µm × 200 µm. The average power delivered to the brain was in the range of 30–120 mW, depending on the depth of imaging.

**Loose-patch recordings under two-photon imaging guidance.** For loose-patch recordings in auditory cortex neurons in vivo, we used the shadow-patching procedure[37,61–63], except that we did not rupture the membrane of targeted cells to maintain a loose-patch configuration. Voltage-clamp recordings were performed with an EPC10 amplifier (HEKA Elektronik, Germany). Patch pipettes were made using a puller (PC-10; Narishige, Tokyo, Japan). The pulling mode was set at two-stage and a heavy type. The glass electrode filled with normal ACSF had a tip resistance of 5–8 MΩ. Applying a pressure of 30 mbar, the glass electrode was lowered and approached the neuron of interest. Once the electrode was moved into the centre of the neuron, we released the pressure and gave a negative pressure (50–100 mbar) until the tip resistance reached 30 M. Raw signals were filtered at 10 kHz and sampled at 20 kHz using Patchmaster software (HEKA Elektronik, Germany).

**Retrograde tracing.** To ascertain that our imaging cortical regions were located in the A1, the criterion that the ventral part of the lateral medial geniculate body (MGBv) connected with A1[64] was used. A glass electrode with a tip diameter of 20–30 µm was filled with neural tracer solution containing Alexa Fluor 488-conjugated cholera toxin subunit B (CTB). Then, we injected the CTB solution by pressure (700 mbar) for 3 min in the imaging site at a depth of 500 µm below the surface. Seven days after the injection, the animals were anaesthetized, and their brains were removed and immersed in 4% paraformaldehyde overnight. A consecutive series of coronal sections (50-µm thick) were collected using a sliding cryotome, and then all sections were mounted onto glass slides and imaged with a stereoscope (Olympus).

**In vivo widefield fluorescence imaging.** A homemade binocular microscope (BM01, SIBET, CAS) with a 4X, 0.2 NA objective (Olympus) was used to record widefield fluorescence images in the mouse cortex for establishing the reference cortical map of A1 (Fig. 2b). A light-emitting diode (470 nm, M470L4, Thorlabs) was used for blue illumination. Green fluorescence passed through a filter cube was measured at 10 Hz with a sCMOS camera (Zyla 4.2, Andor Technology)[7]. Thy1-GCaMP6f mice (Jackson labs stock number 025393, also known as GP5.17) were used to functionally identify the region of A1[65].

The mouse was anaesthetized by urethane and kept on a warm plate (37.5 °C). A piece of bone (~5 mm × 5 mm) (the centre point: Bregma −3.0 mm, 4.5 mm lateral to midline) was removed and replaced by a coverslip (3 mm in diameter, below). To localize A1, four pure tones (4, 8, 16, and 32 kHz) were repeatedly presented 20 times at an interval of 6 s.

In each mouse, the recorded cortical images were first downsampled from the original 750 × 1200 pixels to 75 × 120 pixels. After that, the frames recorded with sound stimuli were averaged across 20 trials. To enhance the signal-to-noise ratio, spatial averaging was conducted over 5 × 5 pixels by a matrix filter, and temporal averaging was conducted with three consecutive images[66]. The pre-processed images were then temporally normalized to obtain the relative changes in fluorescence ($f$) pixel-by-pixel. With the baseline fluorescence ($f_0$) obtained by averaging the images of 800 ms before sound stimulation, the relative fluorescence changes of each pixel were calculated as $\Delta f/f = (f - f_0)/f_0$. The normalized images are shown on a colour-coded scale to visualize the relative fluorescence changes ($\Delta f/f$) in the cortex.

**Data analysis.** We analysed our data using custom-written software in LabVIEW 2012 (National Instruments), Igor Pro 5.0 (Wavemetrics) and MATLAB 2014a (MathWorks)[67]. To correct motion-related artefacts in imaging data, we used a frame-by-frame alignment algorithm to minimize the sum of squared intensity differences between each frame image and a template, which is the average of the selected image frames.

To extract fluorescence signals, we visually identified neurons and performed the drawing of regions of interest (ROIs) based on fluorescence intensity. Fluorescence changes ($f$) were calculated by averaging the corresponding pixel values in each specified ROI. Relative fluorescence changes $\Delta f/f = (f - f_0)/f_0$ were calculated as $Ca^{2+}$ signals, where the baseline fluorescence $f_0$ was estimated as the 25th percentile of the entire fluorescence recording.

As described in our previous studies[26,58], we performed automatic $Ca^{2+}$ transient detection based on thresholding criteria regarding peak amplitude and rising rate[68]. During the licking task, licking activities were semi-automatically tracked from the monitoring movie and quantified as a time course[26,27]. To determine the responding and non-responding neurons, we calculated the $Ca^{2+}$ response amplitude of one spike from the data of the combined two-photon imaging and electrophysiology. The non-responding neurons were defined as the ones with $Ca^{2+}$ response amplitudes significantly smaller than that of one spike.

To compare data (e.g., $Ca^{2+}$ amplitude) between groups, we used non-parametric Wilcoxon rank sum test (unpaired), Wilcoxon signed-rank test (paired)

or bootstrap test to determine statistical significance ($P < 0.05$) between them. To compare data from two groups by bootstrap approach, the original data sets were sampled with replacement 10,000 times, and the $P$-value of the change was computed as the proportion of the bootstrap distribution produced an inconsistent change. One-way analysis of variance (ANOVA) was used to compare averages in multiple conditions. Fisher's exact test and Chi-square test were used to compare two proportions. $P$ values were adjusted for multiple comparisons using the Bonferroni correction. In the text, summarized data are presented as the median/25th–75th percentiles. In the figures, the data presented in the box-and-whisker plot indicate the median (centre line), 25th and 75th percentiles (Q1 and Q3), i.e., interquartile range (IQR) (box), Q1-1.5 × IQR and Q3 + 1.5 × IQR (whiskers), and all other data with error bars are presented as the mean ± s.e.m. (except for Fig. 7, where the variation is presented as ±s.d.).

We used a standard 2-class linear SVM from the built-in functions of the widely used programming platform MATLAB™. A neuronal population response pattern of a trial is defined as a vector consisting of $\Delta f/f$ values of neurons in a defined population. We randomly selected only 1 trial from each class to train the SVM and then randomly selected another 1 trial from each class to test the performance of SVM. For each population size ($N_p$), the accuracy of classification ($A_c$) is the average value (+/−s.d., applies for all values in this section) from 50 different randomly shuffled cell selections, and for each cell selection, we ran 50 different randomly configured training and testing trial sets. The SVM was reset for each iteration of the cell and trial subset configuration. The SVM had no access to cell labels, and only human analysts used the cell labels to display the $A_c$–$N_p$ relation for different recruiting orders.

In the first scenario (as illustrated by Fig. 7b, left cartoon), we used data from mice that were trained with BBN and tested with BBN (target) as well as pure tones of 11 different frequencies (nontarget). For one example imaging FOV, when we began recruiting NB cells into a population, a small number of NB cells could already achieve an accuracy above chance level (Fig. 7b, middle graph), in line with a previous report[69] demonstrating that a few cells may be sufficient to encode a task-related stimulus. Furthermore, the accuracy increased monotonically with increasing $N_p$. When $N_p$ reached the maximum possible number of NB cells (32 in this example experiment), $A_c$ was a remarkable 84.4 ± 0.5%. Strikingly, recruiting 3 HB cells raised $A_c$ to 100% for all selections of training and test data (inset in the middle graph of Fig. 7b). When we repeated the same analysis by recruiting HB cells first, the 3 HB cells were sufficient for achieving perfect discrimination ($A_c = 100\%$), and the additional recruitment of NB cells did not affect $A_c$ (Fig. 7b, right graph). Note that in this particular imaging FOV, there were no qHB or AB cells.

We concatenated the response patterns from all imaging field-of-views in multiple mice (Fig. 7c, left cartoon) and performed the same analysis over such a hypermouse. In the hypermouse, we had all four subtypes of cells present, i.e., NB, AB, qHB and HB cells. Furthermore, random selection of cells tended to mix cells from different mice, disrupting correlations that could contribute to the high discrimination performance. When we recruited the NB cells first (Fig. 7c, middle graph), 614 NB cells provided an $A_c$ of 94.6 ± 0.5%. Interestingly, further recruitment of 37 AB cells reduced $A_c$ to 86 ± 3%. Further recruitment of 8 qHB cells increased the $A_c$ back to 99.6 ± 0.1%, and recruitment of 22 HB cells resulted in 100% $A_c$. Again, initial recruitment of the 30 HB/qHB cells achieved perfect performance ($A_c = 100\%$), and further recruitment of AB and NB cells did not lower the $A_c$ at all (Fig. 7c, right graph).

In the second scenario, we performed similar calculations with responses from mice trained with 2 chords. To maintain simplicity, we continued to use the same 2-class linear SVM trained in the 1-shot learning paradigm. One of the two chords was labelled as the target class, while the other chord and all 8 tones (4 tones in each of the two chords) were labelled as the nontarget class (Fig. 7d, left cartoon). We report the results of these calculations (Fig. 7d, middle and right graphs) in the same format as above. Note that in the 2-chord scenario, the definition of NB and AB cells remained the same as in the BBN scenario, but the HB/qHB cells were further split into 4 subtypes: HB/qHB cells for chord 1 (HB-c1, qHB-c1) and HB/qHB cells for chord 2 (HB-c2, qHB-c2). The definition of HB/qHB cells ensured that all these subtypes did not overlap except for the qHB subtypes; indeed, there was 1 exceptional cell that belonged to both the qHB-c1 and the qHB-c2 subtypes, and we labelled it as qHB-c1 only. Out of 570 cells in the new hypermouse, the number of cells in each of the 6 subtypes was as follows: NB (528), AB (19), qHB-c1 (2), HB-c1 (3), qHB-c2 (3), and HB-c2 (15).

The results were highly similar to those obtained with the hypermouse in the BBN scenario. The $A_c$ level with NB cells recruited first reached 73 ± 4%, but the further recruitment of AB cells pulled the $A_c$ down to chance level (50 ± 5%). Because the nontarget class included not only tones but also a complex sound (chord 1) here, the AB cells that exhibited bursts only to tones nearly completely corrupted the partial information of classification carried by the NB cells. For the same reason, further recruitment of the 2 qHB-c1 and 3 HB-c1 cells did not recover the $A_c$. However, with the 3 qHB-c2 cells recruited, the $A_c$ drastically increased again (to 85 ± 3%). With the last 15 HB-c2 cells recruited, the $A_c$ reached 100% (already reached 100% with 13 HB-c2 cells). In the other way (Fig. 7d, right graph), starting with just one HB-c2 cell already achieved 90 ± 9% $A_c$. Further recruitment of more HB/qHB cells continued to raise the $A_c$ up to 99.94 ± 0.06%. The $A_c$ reached 100% with no variation after further recruitment of 11 AB cells and thereafter remained at 100% with the recruitment of the rest of the AB and NB

cells. In this way, the AB cells could positively contribute to the classification information if the HB/qHB cells were recruited before them.

**Reporting summary**. Further information on research design is available in the Nature Research Reporting Summary linked to this article.

## Data availability
The data that support the findings of this study are available from the corresponding author upon reasonable request. Source data underlying Figs. 1, 3–7 and Supplementary Figs. 2–3 are available as a Source data file. Source data are provided with this paper.

## Code availability
The codes supporting the current study have not been deposited in a public repository, but are available from the corresponding author upon request.

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

## Acknowledgements

The authors are grateful to Dr. X.-Q. Wang, Dr. S. Song, Dr. S. Jacob, Dr. H. Adelsberger, Dr. S. Remy, Dr. O. Barnstedt, M. Fabiszak, Dr. W. Freiwald, Dr. Q. Wen, Dr. D. Kleinfeld, Dr. K. Svoboda, Dr. S. Wu, Dr. R. Quian Quiroga, Dr. J.-D. Wu and Dr. W. Ku for very helpful discussions and comments; to Ms. Jia Lou for help in composing and layout editing of the figures. This study was supported by grants from the National Natural Science Foundation of China (31925018, 31861143038, 31921003, 81671106, 31700933, 81771175, 61705251), the Ministry of Science and Technology of China (2018YFA0109600), the Key Scientific Research Equipment Development Project of the CAS (Super-resolution Microscopy Systems and Key Components, ZDYZ2013-1), and the "100-Talents Program for Elite Engineers" of the CAS (H.J.). A.K. is a Hertie Senior Professor of Neuroscience at TUM and a CAS President's Distinguished International Visiting Fellow. X.C. is a fellow of the CAS Center for Excellence in Brain Science and Intelligence Technology, and also a member of the Institute of Brain and Intelligence at the Third Military Medical University.

## Author contributions

X.C., I.N. and H.J. conceived the project. X.C., I.N., M.W., R.L., R.D., S.Z. and X. Liao designed the experiments; M.W., R.L., R.D., J.-C.L., J.Z., W.H., K.L., J.P., Z. Zhao., T.L., K.Z., X. Li, Z. Zhou, J.L., H.J., X.C., Y.Z. and J.Y. performed the experiments; X.C., H.J., I.N., X. Liao, A.K., J.-K.L., Z.V. and Y.M. devised the data analysis methods; X. Liao, M.W., S.L., R.L., I.N., H.J. and X.C. performed the data analysis; H.J., I.N., A.K. and X.C. inspected the data and evaluated the findings; H.J., I.N. and X.C. wrote the manuscript with the help from all authors.

## Competing interests

The authors declare no competing interests.
