## [Peer Review File · Nature Communications]

Reviewers' Comments:

Reviewer #1:

Remarks to the Author:

The authors have satisfactorily addressed my concerns.

But I suggest that the manuscript be further edited by someone for whom English is their native language.

Reviewer #2:

Remarks to the Author:

This is a technically challenging study, which combines two-photon Ca imaging and loose-patch recordings. The authors concluded that learning induced firing pattern changes in a very sparse number of A1 L2/3 neurons which become bursting firing after training. The findings of study are potentially interesting. However, there are some concerns about the main points which are not well supported by the results, and some data are not consistent with the literature. A few technical issues also need to be addressed by additional analyses and data.

Specific comments:

1. The title does not reflect the content of the study which is about behavioral training induced plasticity in A1. Heterogeneity of A1 neurons are well documented in their response specificity and firing patterns, including bursting firing neurons. The title is thus confusing.
2. The impression on the use of "holistic" here is quite vague, or scientifically not rigorous. Regarding fig.6cd, it's known that a large population of A1 neurons only responds to noise instead of tone pips even in naive animals. In addition, for chord test in fig.6ab, only individual tone component was tested, which is too limited to claim that the chord will generate the strongest responses as other combinations of tones could do better. These quantification doesn't provide evidence to support the use of the term. Unless it can be well justified, it'd be better to remove it.
3. The bursting response data are confusing and problematic, which are not consistent with the literature. The bursting responses are defined by more than 3 spikes elicited by a 50 ms BBN in a ~200 ms time window following the start of the sound. There are a few issues, first, how sound evoked spikes are defined is not clear. Normally, it is defined for onset and sustained responses, but not including off-responses. Here for the given time window, the off responses are also included. What does that mean for the behavioral training and for the response change? This analysis needs to be reconsidered.
Second, in awake A1, only a small portion of cells exhibits one-spike onset responses elicited by tone or noise stimulus. In another words, bursting neurons should be common in A1 if not the majority, so it's quite weird for the 2% and 7% of bursting neurons observed in this study. Could this be due to the damage of the cortex given the whole challenge processes?
Thirdly, sample size for loose-patch is too small. The potential bias is a concern. A distribution of the spike responses from different cells will be critical for any claim of response types in the study.
4. Besides the bursting cells, the overall increase of responses in A1 cells, although small, could be also required or sufficient for the learned discrimination task. This needs to be tested. The conclusion should be more balanced instead of simply focusing on the potential contribution of those bursting cells.

We would like to express our deepest appreciation to the reviewers for their constructive comments on our manuscript entitled “‘Holistic bursting’ cells in auditory cortex” (NCOMMS-20-04670-T). To address the reviewers’ comments, we have performed new analyses and revised the manuscript accordingly. We marked the important content changes in the manuscript in red color (language editing was not marked), and here is a summary of them:

- (1) We changed our manuscript title to “‘Holistic Bursting’ Cells Emerge in Auditory Cortex of Trained Mice” based on reviewer #2’s comments.
- (2) We provided a detailed definition of ‘burst firing’ in our data, based on reviewer #2’s comments on our loose-patch recording data.
- (3) We provided a new analysis showing that, in the identified ‘holistic bursting’ cells, the preferred chord-evoked response was greater than the summation of responses evoked by 4 constituent tones (Figure 6g).
- (4) We modified some descriptions regarding Figure 7 such that the conclusions are now more balanced as requested by reviewer #2.

Reviewer's comments

Reviewer #1 (Remarks to the Author):

The authors have satisfactorily addressed my concerns.

But I suggest that the manuscript be further edited by someone for whom English is their native language.

We would like to thank the reviewer for important and helpful comments and suggestions for improving our manuscript. Following the suggestion here, we now have used the Nature-Springer language editing service to improve the English writing (please see the attached certificate below)

SPRINGER NATURE

Author Services

Editing Certificate

This document certifies that the manuscript
'Holistic bursting' cells emerged in auditory cortex of trained mice

prepared by the authors

Meng Wang, Xiang Liao, Ruijie Li, Shanshan Liang, Ran Ding, Jingcheng Li, Jianxiong Zhang, Wenjing He, Ke Liu, Junxia Pan, Zhikai Zhao, Tong Li,...

was edited for proper English language, grammar, punctuation, spelling, and overall style by one or more of the highly qualified native English speaking editors at SNAS.

This certificate was issued on **April 12, 2020** and may be verified on the SNAS website using the verification code **403A-83E4-E8A1-B03E-B6CP**.

Neither the research content nor the authors' intentions were altered in any way during the editing process. Documents receiving this certification should be English-ready for publication; however, the author has the ability to accept or reject our suggestions and changes. To verify the final SNAS edited version, please visit our verification page at secure.authorservices.springernature.com/certificate/verify. If you have any questions or concerns about this edited document, please contact SNAS at support@as.springernature.com.

SNAS provides a range of editing, translation, and manuscript services for researchers and publishers around the world. For more information about our company, services, and partner discounts, please visit authorservices.springernature.com.

Reviewer #2 (Remarks to the Author):

This is a technically challenging study, which combines two-photon Ca imaging and loose-patch recordings. The authors concluded that learning induced firing pattern changes in a very sparse number of A1 L2/3 neurons which become bursting firing after training. The findings of study are potentially interesting. However, there are some concerns about the main points which are not well supported by the results, and some data are not consistent with the literature. A few technical issues also need to be addressed by additional analyses and data.

We would like to thank the reviewer for important and helpful comments and suggestions for improving our manuscript. In the revised manuscript, we have addressed the reviewer's concern by providing additional data analysis as well as rewording some descriptions.

Specific comments:

1. The title does not reflect the content of the study which is about behavioral training induced plasticity in A1. Heterogeneity of A1 neurons are well documented in their response specificity and firing patterns, including bursting firing neurons. The title is thus confusing.

We agree that the title did not fully represent the content of the study. The new title is "Holistic' Bursting Cells Emerge in Auditory Cortex of Trained Mice".

2. The impression on the use of "holistic" here is quite vague, or scientifically not rigorous. Regarding fig.6cd, it's known that a large population of A1 neurons only responds to noise instead of tone pips even in naive animals. In addition, for chord test in fig.6ab, only individual tone component was tested, which is too limited to claim that the chord will generate the strongest responses as other combinations of tones could do better. These quantification doesn't provide evidence to support the use of the term. Unless it can be well justified, it'd be better to remove it.

Following the reviewer's suggestion, in order to better support the use of term 'holistic', we performed an additional analysis for the 'holistic bursting' cells that were identified by the thresholding criteria (i.e., with trial-averaged Ca^{2+} response amplitude $\Delta f/f$ (Cal-520) greater than 1.5). The result showed that the preferred chord-evoked response was not just greater than a single tone-evoked response, but even greater than the arithmetic summation of 4 responses evoked by 4 constituent tones. These new results are shown now in Figure 6g.

3. The bursting response data are confusing and problematic, which are not consistent with the literature. The bursting responses are defined by more than 3 spikes elicited by a 50 ms BBN in a ~200 ms time window following the start of the sound. There are a few issues, first, how sound evoked spikes are defined is not clear.

Thanks to the reviewer's comment. We now give a detailed definition of bursting spike responses: sound-triggered responses consisting of 3 or more spikes in a 150-ms time window from the stimulus onset (equivalent to 20 Hz instantaneous firing rate) and with the presence of spike waveform amplitude decaying in the same time window. This newly added definition does not change our result and conclusion, as the burst response events recorded in our loose-patch data were typically with instantaneous firing rates that were even much higher (Fig. 3f, inter-spike-interval time 11 / 8.5–21 ms, corresponding to firing rate of 90 / 50 - 120 Hz).

Normally, it is defined for onset and sustained responses, but not including off-responses. Here for the given time window, the off responses are also included. What does that mean for the behavioral training and for the response change? This analysis needs to be reconsidered.

To address this question, we pooled the spike responses from all loose-patch recorded neurons for both groups of naïve and trained animals (see the data plot below).

In neither of these datasets we could observe a clear second-peak response after the end of sound stimulus (50 ms), indicating the absence of off-responses or the overlapping of on- and off-responses in

our results. This is well consistent with a previous study from the Michael Wehr group showing that off-responses could be evoked by sound stimuli with a duration of longer than 100 ms in rat auditory cortex (Ben Scholl et al, Neuron, 2010). Similarly, in the auditory cortex of cats, off-responses obviously decreased and overlapped with on-responses when the sound stimulus duration was shorter than 40 ms (Qin et al., J Neurosci, 2009) or 50 ms (He et al., J Neurosci, 1997). Thus, a segregated analysis for on-/off- responses was not appropriate for our study.

Second, in awake A1, only a small portion of cells exhibits one-spike onset responses elicited by tone or noise stimulus. In another words, bursting neurons should be common in A1 if not the majority, so it's quite weird for the 2% and 7% of bursting neurons observed in this study. Could this be due to the damage of the cortex given the whole challenge processes?

Throughout our study, we had been always careful with the recording quality. We did not chronically implant microelectrodes and did not insert microelectrodes too often (maximally 3 cells recorded per animal, a protocol for patch-clamp recordings *in vivo* we described previously (Chen et al., Nature Protocols, 2012)), so the damage to the cortex was not a major concern.

For loose-patch recordings, we did not sample cells randomly, but we specifically targeted cells by selecting from the online readout of Ca^{2+} response amplitudes (Fig. 3a, b). Our targeting was made on purpose, first to record those few cells with high Ca^{2+} responses (and revealed burst firing with the spike waveform amplitude decaying, Fig. 3c, d), then to record more cells that evenly covered the response dynamic range in order to establish the relation between Ca^{2+} response and spiking response ('calibration graph', Fig. 3i). Thus, our loose-patch data shall not be interpreted in terms of population distribution (i.e., Fig. 3e is only a summary of sampled cells but cannot represent the distribution of the entire A1 L2/3 population).

On the other hand, our Ca^{2+} imaging data with a massive sampling of thousands of cells clearly showed that the population distribution of responses was highly skewed (see the log-normal distribution plots in Fig. 4). Our estimate of the bursting responsive cell ratio in naïve animals was 2.1% (9/423) with GCaMP6f imaging or 0.2% (2/1314) with Cal-520 imaging; and in trained animals it was 7.1% (30/423) with GCaMP6f imaging or 4.7% (52/1112) with Cal-520 imaging. Dr. Anthony Zador's lab has performed a major quantitative analysis of sound responses in the awake (naïve) rodent auditory cortex by means of blind loose-patch recording, and found that sound stimuli elicited high firing rates (≥ 20 Hz, equivalent to our new definition of bursts) in less than 5% of neurons at any instant (Hromadka et al., PLOS Biology, 2008). Assuming the methodological differences (i.e., blind patching has a potential

sampling bias towards highly responsive cells), our results are consistent with those in the Dr. Zador's paper.

Thirdly, sample size for loose-patch is too small. The potential bias is a concern. A distribution of the spike responses from different cells will be critical for any claim of response types in the study.

As mentioned in the previous point, our loose-patch recording was made on purpose with imaging-guided targeting. However, our conclusions were drawn from the results of the combination of loose-patch recording and two-photon Ca^{2+} imaging. On one hand, with loose-patch recordings, we established a full-range calibration of Ca^{2+} response (Fig. 3i, Suppl. Fig. 4) by recording the spike firing responses simultaneously with the Ca^{2+} responses for a subset of cells. On the other hand, with extended datasets of Ca^{2+} imaging, beyond the calibration experiments, thousands of cells were sampled in naïve and trained animals (Fig. 4). The highly skewed, log-normal shaped population distributions of responses clearly showed that the highly responsive (bursting) cells were indeed the minority.

4. Besides the bursting cells, the overall increase of responses in A1 cells, although small, could be also required or sufficient for the learned discrimination task. This needs to be tested.

We analyzed the overall population responses as requested by the reviewer, and the result is shown in the figure here below:

This figure shows that the responses of non-bursting (NB) population exhibited nearly the same level of responses to the trained complex sound(s) as to the constituent tones ($p > 0.05$ for any relevant paired rank sum test; the two right panels: the upper one for BBN-trained mice, and the lower one for 2-chord trained mice). Thus, if we only consider the overall responses, the NB cells could be neither required nor sufficient for the discrimination task. We prefer to not include this insignificant test result in the manuscript, because with the SVM test (Fig. 7) we were able to show how the NB cells could contribute to the population information coding.

The conclusion should be more balanced instead of simply focusing on the potential contribution of those bursting cells.

In the description of Figure 7, we modified some words in the text such that the conclusion is now more balanced. The new conclusion states that linear decoding of a large set of NB cells could achieve a significant discrimination accuracy of single trials, but the very few HB/qHB cells could achieve a perfect discrimination under the same conditions.

Reviewers' Comments:

Reviewer #2:

Remarks to the Author:

Authors addressed some of the technical issues raised in the revision. However, besides some challenging technical approaches, the impression is more about that the phenomenological study doesn't really generate any new insights. The take-home message, behavioral training can strengthen cortical responses (even some bursting cells), is well documented in the literature.

Argument for 'holistic' based on supralinear summation is still mostly speculative. The term is not well supported as the major conclusion of the paper, but rather a topic for discussion.

The reply also raises another question. L2/3 of A1 is well known for sparse coding, but 100% of neurons imaged in this study seem to respond to tones/BBN. Why?

We would like to express our deep appreciation to the reviewer for his/her constructive suggestions. Here we provide a point-by-point response to the reviewer comments.

REVIEWERS' COMMENTS:

Reviewer #2 (Remarks to the Author):

Authors addressed some of the technical issues raised in the revision. However, besides some challenging technical approaches, the impression is more about that the phenomenological study doesn't really generate any new insights. The take-home message, behavioral training can strengthen cortical responses (even some bursting cells), is well documented in the literature. Argument for 'holistic' based on supralinear summation is still mostly speculative. The term is not well supported as the major conclusion of the paper, but rather a topic for discussion. The reply also raises another question. L2/3 of A1 is well known for sparse coding, but 100% of neurons imaged in this study seem to respond to tones/BBN. Why?

We now avoided using the phrase holistic bursting either in the Title or the Abstract. In addition, we introduced the phrase holistic bursting in the discussion as well as in the results accordingly (as changes tracked). The term holistic bursting is now used as a note to help describing data but not as a conclusive statement.

Regarding the new question that "100% of neurons imaged in this study seem to respond to tones/BBN", we think that it was a misunderstanding. For example, In figure 2f, most neurons did not respond to BBN (blue dots). In figure 5b,d, most example neurons (cell 3,4,7,8,9) showed no response to BBN. The example neurons were chosen to be more closely representing the actual distribution.

The misunderstanding could have come from figure 6 in which indeed all the shown neurons were with strong responses (with Ca^{2+} signal amplitude $df/f > 1.5$, corresponding to burst firing response) to either BBN or chord or tones, however, it was explicitly written that these neurons were identified out of a large population of all imaged neurons, whereas the rest majority neuron did not show such strong responses to any stimulus and even many neurons did not respond to any stimulus at all. The exact numbers of each kind of neurons as well as the total number of imaged neurons in each dataset were given in the main text.

In addition, based on the analysis of single-trial Ca^{2+} response amplitude and the number of spikes per response event (Cal-520: Fig. 3j; GCaMP6f: Supplementary Fig. 2c), the Ca^{2+} response amplitudes for one spike were ~ 0.41 df/f (mean, $n = 20$ single-spike-associated Ca^{2+} responses) and ~ 0.42 df/f (mean, $n = 27$ single-spike-associated Ca^{2+} responses) for Cal-520 and GCaMP6f, respectively. Therefore, the nonresponding neurons were defined as the ones

with Ca^{2+} response amplitudes significantly smaller than that of one spike. According to this definition, in Figure 4, a large number of neurons were nonresponding ones upon BBN stimulation.